# Mucosal immune responses and intestinal microbiome associations in wild spotted hyenas (*Crocuta crocuta*)
Susana P. Veloso Soares [1,2] ✉, Victor H. Jarquín-Díaz [3,4,11], Miguel M. Veiga[1,2], Stephan Karl[1], Gábor Á. Czirják [2], Alexandra Weyrich [5,6], Sonja Metzger[1], Marion L. East[1], Heribert Hofer [7,8,9], Emanuel Heitlinger[3,4], Sarah Benhaiem [1] ✉ & Susana C. M. Ferreira [10] ✉

Little is known about host-gut microbiome interactions within natural populations at the intestinal mucosa, the primary interface. We investigate associations between the intestinal microbiome and mucosal immune measures while controlling for host, social and ecological factors in 199 samples of 158 wild spotted hyenas (*Crocuta crocuta*) in the Serengeti National Park, Tanzania. We profile the microbiome composition using a multi-amplicon approach and measure faecal immunoglobulin A and mucin. Probabilistic models indicate that both immune measures predicted microbiome similarity among individuals in an age-dependent manner. These associations are the strongest within bacteria, intermediate within parasites, and weakest within fungi communities. Machine learning models accurately predicted both immune measures and identify the taxa driving these associations: symbiotic bacteria reported in humans and laboratory mice, unclassified bacteria, parasitic hookworms and fungi. These findings improve our understanding of the gut microbiome, its drivers, and interactions in wild populations under natural selection.

The gut microbiome - the community of micro-organisms and macro-organisms and their products or genetic material that intimately reside within the gastrointestinal tract - can influence host physiology and plays a vital role in maintaining host homeostasis and health[1,2]. In turn, the host's physiology, particularly the mucosal immune system, shapes this community. The intestinal epithelium, a complex layered set of cells covered by a mucus layer, is the first line of defence against pathogen translocation and also fosters the maintenance of beneficial taxa[3,4]. It produces and releases a variety of enzymes and immune defence molecules such as antimicrobial peptides, mucins, and antibodies, to maintain intestinal homeostasis[5–7]. Perturbations of this community can be associated with intestinal diseases, often accompanied by increased mucosal permeability and disrupted immune responses[8–10].

Two important and broad-acting measures of mucosal immunity are secretory immunoglobulin A (IgA) and mucins. IgA is the primary antibody secreted into the intestinal lumen. It coats the surface of a broad but defined subset of gastrointestinal taxa[11] and is a substantial metabolic substrate for microorganisms[4,12]. IgA also prevents pathogens from crossing the intestinal epithelium and neutralises toxins and virulence factors[13]. These processes selectively promote or hamper the colonisation and growth of specific taxa, thereby supporting the intestinal barrier in the control of the gut microbiome. Mucins form a mucus layer that covers the epithelium, serves as a lubricant to protect against mechanical stress, and limits its direct contact with toxins, digestive enzymes, and potential pathogens[14,15]. Mucins are also important metabolic substrates for the gut microbiome, providing attachment sites that promote colonization and provide a selective environment for symbionts[15,16]. Despite a growing body of knowledge of interactions between intestinal mucosal immunity and the microbiome, most studies are focused on bacteria[17,18].

Although bacteria outnumber eukaryotes within mammalian intestines[19], eukaryotes such as protozoa, helminths, and fungi also play a significant role in regulating the gut microbiome[20] and host health[21–23]. Studies on immune responses to parasites (i.e. protozoans and helminths) and fungi are often performed in isolation from the rest of the intestinal community, although interactions with other taxa likely shape both the gut microbiome and immune responses[24–26]. Host-microbiome interactions via mucosal immunity are poorly understood, and this applies even more so to wild animal populations where the acquisition of samples can be challenging and there are few species-specific reagents and appropriate immune assay validations[27].

Wild animals living in natural environments can have a distinct microbiome compared to their captive counterparts[28–32], with a higher diversity of eukaryotes[33,34]. The study of host-microbiome interactions in wild animals has also provided important insights into how host characteristics and the ecological environment shape their intestinal communities. Gut microbiome composition changes with age and between life

---

stages, particularly during ontogeny, as observed in several mammals[35–37]. This effect is often attributed to changes in host physiology, particularly immune development, but sometimes also to behavioural and dietary changes. Additionally, the host environment (biotic and abiotic factors) shapes the composition of gut microbiomes, as seen by spatial and temporal heterogeneities[38–40], the effect of diet[41,42] and social interactions[43,44]. Importantly, the environment interacts with host physiology and genetics, and these interactions might change over time and are as such context dependent[45]. Long-term field studies allow the collection of detailed data on the life histories of individually recognised animals and on the biotic and abiotic environment, with minimal anthropogenic manipulation. These are particularly suitable to disentangle the relative contributions of host characteristics and ecological factors in shaping microbiome composition and potential interactions within this community and with the host[46–48].

Our study aims to investigate the links between intestinal mucosal immunity and the gut microbiome, using data and samples from a long-term research project on wild spotted hyenas (*Crocuta crocuta*) in the Serengeti National Park (SNP). Spotted hyenas (hereafter 'hyenas') live in stable social groups (clans) with members varying in their social status within a linear dominance hierarchy, in which adult females and their offspring socially dominate immigrant males[49–51]. Because they hunt and scavenge and thus can consume fresh and highly decomposed carcasses[52], hyenas can be exposed to a variety of pathogens. Additionally, pathogen transmission within clan members is facilitated by their high rate of social interactions[53–58]. Nonetheless, hyenas are known to remain healthy[53,59], suggesting specific adaptations such as a specialised immune system and/or resilient intestinal community[54,60]. Previous research on Serengeti hyenas has shown that host characteristics such as age and the social environment, including clan identity and structure, affect parasite infections[57,61,62] and gut bacterial composition[35,63,64]. Age is a particularly important factor, as it affects parasite composition[57,61,62], gut bacterial composition and modulates faecal IgA and mucin[65]. Furthermore, faecal IgA and mucin reflect *Ancylostoma* faecal egg load, a parasite shown to reduce juvenile survival and longevity in the Serengeti population[57,65].

Here, we hypothesise that an individual's gut microbiome is regulated by its mucosa immune responses, including IgA and mucin levels. We profiled the intestinal community (overall microbiome) and its components, bacteria, parasites and fungi. We tested whether 1) mucosal immune measures (faecal IgA and mucin) were associated with gut microbiome composition, whether these relationships were 2) modulated by host age, and whether 3) broad associations exist among the microbiome components (bacteria, parasites and fungi), whilst accounting for other abiotic and social environment factors relevant to the individual host. We further explored inter-species interactions among the gut microbiome, and highlighted the taxa with the strongest links to immune measures. Our findings indicate that the gut microbiome is shaped by host and environmental factors such as age at and time of sampling, and is strongly associated with mucosal immune responses. Mucosal immune measures had strong associations with specific taxa of bacteria, parasites and fungi known to establish mutualistic, commensal and parasitic interactions. We also found strong associations among members of the gut microbiome.

## Materials and methods
### Study site, population under study, and sample collection
We collected life-history, behavioural and ecological data, and 210 faecal samples non-invasively from 165 individually recognised wild spotted hyenas from 2004 to 2018. To conduct fieldwork in Tanzania we were granted annual research permits from the Tanzania Commission for Science and Technology (COSTECH) and permission from the Tanzanian National Parks Authority (TANAPA) and Tanzanian Wildlife Research Institute (TAWIRI). All animal procedures and protocols were approved by the Leibniz Institute for Zoo and Wildlife Research Ethics Committee on Animal Welfare (permit number: 2017-11-02). We have complied with all relevant ethical regulations for animal use. Our samples were exported in several batches. All the necessary permits were obtained for exporting these samples.

Hyenas from this study belong to three clans monitored on a near daily basis since 1987, 1989, and 1990 in the context of an ongoing long-term individual-based research project in the Serengeti National Park, Tanzania. Individuals are recognised based on their unique spot patterns and other features, such as scars and ear notches[49,66]. We limited our focus to cubs of both sexes and adult females as most adult males tend to disperse upon reaching adulthood, making it challenging to monitor them throughout their entire lifespan[61].

Cubs were aged to an accuracy of ±7 days based on their behaviour, movement coordination, size and pelage when seen for the first time[67]. By the age of approximately three months, sex was determined by the shape of the external genitalia, particularly the dimorphic glans morphology of the erect phallus[68]. Maternal identity was determined based on nursing observations at the communal den(s) and was further confirmed by DNA microsatellite loci analysis[69]. Approximately 1250 hyenas were genotyped at 9 microsatellite loci (see in ref. 70) representing about 41% of the individuals born into one of the three study clans since the start of the project in 1987.

The social rank of adult females in their clans was determined based on the observation of submissive acts in dyadic interactions recorded *ad libitum* and during focal observations[71]. For each clan, we used the outcome of these dyadic interactions to construct an adult female linear dominance hierarchy that was updated daily after demographic changes (recruitment or deaths of adult females) and socially mediated changes in rank (coups). To make rank positions comparable across clans and within clans when the number of females in the hierarchy changed, we assigned standardised ranks, evenly distributing social ranks from the highest (standardised rank: +1) to the lowest rank (standardised rank: −1) within a clan. Juveniles (individuals younger than two years) were assigned the same standardised ranks as the mothers raising them (typically their genetic mother) at the sampling date[72]. Six hyenas sampled before reaching adulthood were adopted or jointly raised by both their genetic and surrogate mothers[72]. In these cases, we assigned the social rank of the surrogate mother or the average of the rank of the mothers in the case of joint-raising.

Non-invasive faecal samples were opportunistically and immediately collected after defecation. The researchers used the same research vehicle to which animals are habituated and were careful not to be seen when collecting the samples, by positioning the vehicle very close to the faeces, to guarantee the safety of the researchers and minimise disturbance to the animals. Samples were collected in individual labelled bags and refrigerated in cool boxes with frozen ice packs in the field until transport to the field station (no more than 3–4 h later). At the field station, they were mechanically mixed and aliquots stored at −10 °C until their transport to storage at − 80 °C at the IZW[61]. Aliquots for DNA extraction were stored in RNAlater (Sigma-Aldrich, St Louis, MO, USA).

### Faecal immunological assays
Faeces aliquots were lyophilised for 22 h in a freeze-dryer (Epsilon1-4 LSCplus, Martin Christ GmbH, Osterode, Germany) followed by homogenisation with mortar and pestles. Faecal mucin (f-mucin) and faecal immunoglobulin A (f-IgA) assays, previously adapted and validated for application to spotted hyena faeces, were applied using the methods described in detail in ref. 65. The faecal mucin assay detects and quantifies oligosaccharides released from mucin by discriminating between O-linked and N-linked glycoproteins[73,74] in freeze-dried samples. Briefly, freeze-dried samples were suspended in phosphate-buffered saline (PBS) solution and then mixed and incubated in a shaking bath to denature glycosidases. Incubation was followed by centrifugation for solubilisation and separation. The resulting supernatant was mixed with alkaline 2-cyanoacetamide (CNA), and incubated. Borate buffer was added, and the solution was cooled to room temperature. The measurements were conducted using a fluorometric assay at 383 nm with an excitatory wavelength of 336 nm (Infinite M200, TECAN, Männedorf, Switzerland), with a low-binding 96-well plate (PerkinElmer(R)). Results are expressed as μmol oligosaccharide equivalents to a standard curve created with N-acetylgalactosamine (Sigma-Aldrich(R), Darmstadt, Germany), whereas porcine stomach mucin was

used as a positive control (Sigma-Aldrich(R), Darmstadt, Germany). f-IgA levels were measured using a sandwich ELISA altered from in ref. 75. Briefly, saline extracts from freeze-dried faecal samples were used[76,77], and protease-inhibitor MixM (Serva Electrophoresis GmbH, Heidelberg, Germany) was added. The resulting supernatant was stored at −20 °C until use. For the sandwich ELISA, an anti-cat IgA (Lot.A10, Novusbio, Abingdon, UK) was used as the capture antibody, and conjugated anti-cat IgA (Lot.P18, Novusbio, Abingdon, UK) as detection antibody. The plates were read at a wavelength of 450 nm using the BioTek Quant Microplate reader (BioTek, Vermont, USA). The results are presented as relative units (RU), and standard curves were obtained with a pool of 72 samples. For both mucin and f-IgA assays, all plates included negative controls and quality controls, and all samples were performed in duplicate and results were accepted if the coefficient of variation was below 5% and within the working range previously established[65].

### DNA extraction of faecal samples for microbiome assessment

We extracted genomic DNA from faeces using the NucleoSpin®Soil kit (Macherey-Nagel GmbH & Co. KG, Düren, Germany) under the manufacturer's protocol with the following modifications: we performed mechanical lysis of the sample in a Precellys®24 high-speed benchtop homogeniser (Bertin Technologies, Aix-en-Provence, France) with two disruption cycles at 6000 rpm for 30 s, with a 15 s delay between them. We eluted DNA in a 40 µL TE buffer. Quality and integrity of the DNA were measured with a full-spectrum spectrophotometer (NanoDrop 2000c; Thermo Fisher Scientific, Waltham, MA USA). We used a Qubit® Fluorometer and the dsDNA BR (broad-range) Assay Kit (Thermo Fisher Scientific) to quantify the concentrations of double-stranded DNA. The DNA extracts were adjusted to a final concentration of 50 ng/µl with nuclease-free water (Carl-Roth GmbH + Co. KG), and stored at −80 °C until further use. The NucleoSpin®Soil kit has previously been successful in uncovering bacteria and eukaryotes, including the parasitic phyla apicomplexa, nematoda and platyhelminthes, in faecal samples from spotted hyenas[35].

### Library preparation and sequencing

We used faecal DNA preparations for multimarker amplification using the microfluidics PCR system Fluidigm Access Array 48 × 48 (Fluidigm, San Francisco, California, USA). We randomised the sample order and amplified individuals' DNA in parallel with non-template negative controls using a microfluidics PCR. This allows a broad-range amplification of multiple fragments (amplicons) for prokaryotic and eukaryotic ribosomal genes (16S, 18S, 28S), intergenic regions (ITS1, ITS2), mitochondrial genes (COI, 16S, CytB), apicoplast genes (tRNA, ORF470) and other protein encoding nuclear genes (HSP70). The list of primer pairs, target genes and regions are described in the Supplementary Data 1. We integrated library preparation into the amplification PCR setup according to the protocol for Access Array Barcode Library for Illumina Sequencers (single direction indexing) as described by the manufacturer (Fluidigm, San Francisco, California, USA). The amplicons were quantified using the Qubit fluorometric quantification dsDNA High Sensitivity Kit (Thermo Fisher Scientific, Waltham, USA) and pooled in equimolar concentrations. The final library was purified using Agencourt AMPure XP Reagent beads (Beckman Coulter Life Sciences, Krefeld, Germany). The quality and integrity of the library were evaluated using the Agilent 2200 TapeStation with D1000 ScreenTapes (Agilent Technologies, Santa Clara, California, USA). Sequences were generated at the Berlin Center for Genomics in Biodiversity Research (BeGenDiv) on the Illumina MiSeq platform (Illumina, San Diego, California, USA) using v2 chemistry with 500 cycles. All sequences are accessible in BioProject PRJNA1134446 in the NCBI Short Read Archive (SRA).

### Identification and quality screening of the amplicon sequence variants (ASVs)

All analyses were performed using R v 4.4.0[78]. For the initial analysis we used the packages dada2 v. 4.3.1[79] and MultiAmplicon v. 0.1.1[80] to filter, sort, merge, denoise, and remove chimaeras for each run separately and for each amplicon, and removed 34 contaminant ASVs with decontam v. 1.21.0[81] using "prevalence" and "frequency" methods (method = "combined") based on DNA concentration estimated spectrophotometrically with NanoDrop (Thermo Fisher Scientific, Germany). We further removed amplicon sequence variants (ASVs) present in only one sample and samples with less than 100 reads (Supplementary Fig. 2). Each amplicon in the multi-amplicon datasets was individually filtered and normalised by total sum scaling for relative abundances and then all products were collated into an "phyloseq" object with the function "merge_phyloseq" from the package phyloseq v. 1.45.0[82] in R. This last step resulted in 199 samples from 158 individuals. An overview of the entire bioinformatic pipeline is shown in Supplementary Fig. 1, section Supplementary Figs. on supplements file.

### Taxonomic annotation of ASVs

We used the RDP classifier[83] implemented within the package dada2 v. 1.29.0 in R[79], to assign taxonomy to the resulting ASVs. We used the SILVA 138.1 SSU Ref NR 99 to classify eukaryotic 18S and bacterial 16S rRNA gene sequences, the SILVA 138.1 LSU Ref NR 99 databases[84] for eukaryotic 28S rRNA gene sequences, and the UNITE database[85] for eukaryotic ITS rRNA gene sequences. All other sequences from targeted regions without publicly available curated databases were classified against sequences downloaded from NCBI using RESCRIPt[86].

### Merging ASVs into combined ASV (cASV)

With the multi-amplicon approach we target different marker loci of the same taxon, resulting in multiple ASVs from the same taxon. For this reason, we identified ASVs likely originating from the same taxon on the basis of their co-abundance within each genus ($n = 476$) and merged them into one combined ASV (cASV) as illustrated in Supplementary Fig. 1B - supplements file. Co-abundance networks were constructed based on positive (Pearson coefficient >0) and significant correlations (p < 0.01), after correcting for multiple testing with the Benjamini-Hochberg method. ASVs that clustered together using the "cluster_fast_greedy" function from the package igraph v. 2.0.3[87] in R were then merged into one ASV by summing their abundances into cASV. This has been previously accessed for Eimeria spp.[40,88] and Oxyuridae, and extended to multiple microbial targets[40]. The resulting ASV/cASV table contains ASVs that did not cluster with any other ASV and cASVs. For simplification purposes, we use the term cASV regardless whether it refers to a single or combined ASVs.

### Statistical analysis and reproducibility

We investigated the intestinal community by considering all taxa detected in the samples, regardless of the overall taxonomic annotation, and by decomposing the community into three components of cASVs: 1) bacteria domain, 2) eukaryotic parasites, and 3) fungi kingdom. We investigated the gut microbiome variation among samples by calculating the pairwise distances (β-diversity) based on the abundance (Bray-Curtis distances) of all identified cASVs, using the R package vegan v. 2.6-4[89] with the function "distance". Dissimilarity distances were then transposed to similarity distances (1-Bray-Curtis distances).

We tested the effect of individuality of microbiome phenotypes (estimated as repeatability through time) on ß-diversity measures in 78 samples from 37 individuals sampled 2 to 3 times, with a mean sampling per individual of 2, in the overall microbiome and in restricted datasets of bacteria, parasite, and fungi members. We used distance-based intraclass correlation coefficients (dICC) and standard errors (SE) calculated based on 1000 bootstrap iterations, implemented with the R package GUniFrac v.1.7[90].

We tested for the association between immune measures and the β-diversity of the intestinal microbiome composition (microbiome similarity among compared samples), while accounting for the effects of other known or expected host, social, and ecological variables, using pairwise comparisons (distances) among samples (excluding within-sample comparisons), as previously described[44,88,91]. Host variables included age difference in days, distances in f-IgA, f-mucin levels and the genetic mother (same vs different). The ecological and social environment variables included the standardised

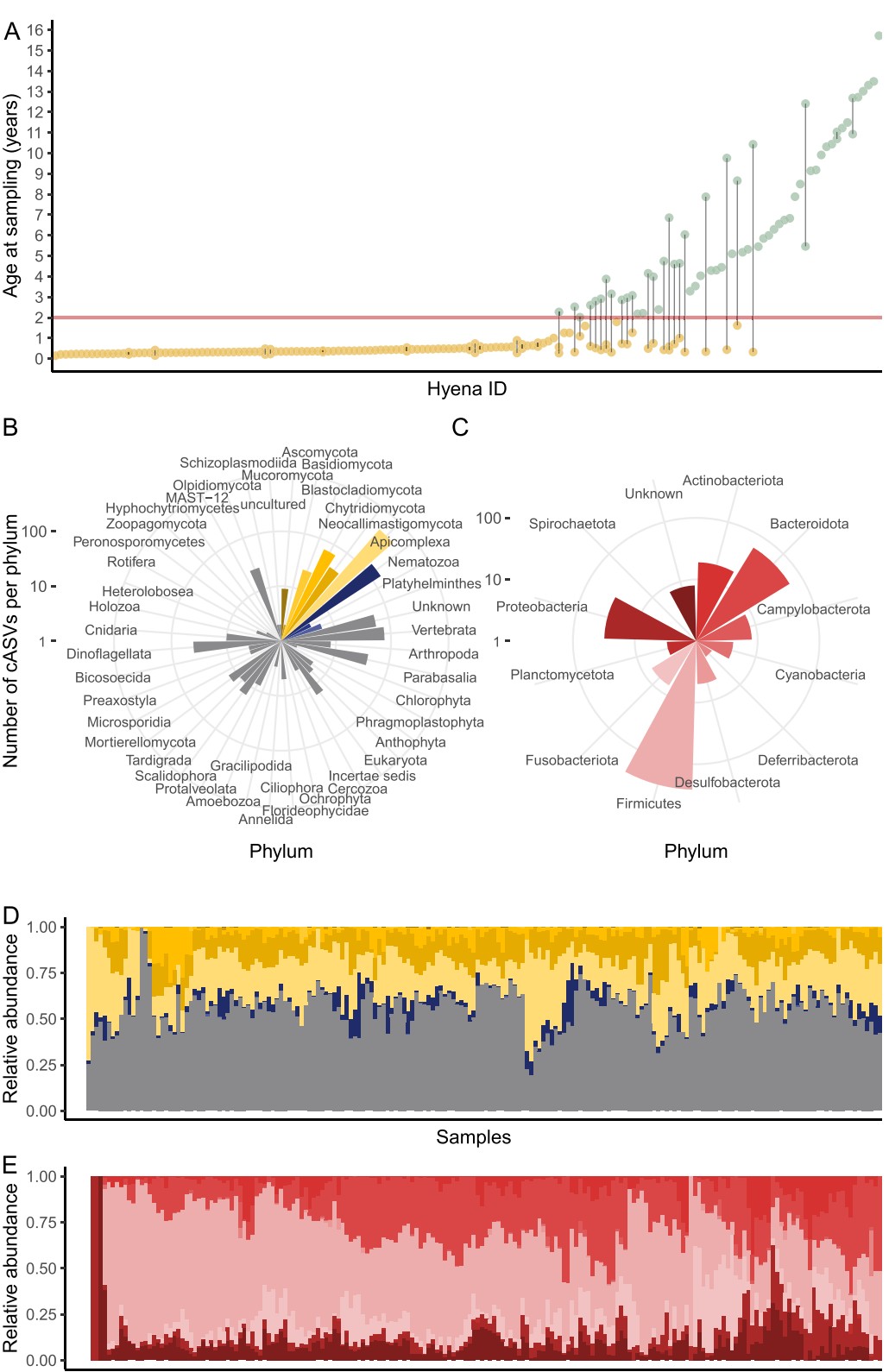

**Fig. 1 | The intestinal microbiome composition of juvenile and adult spotted hyenas. A** Distribution of age at sampling. Samples from the same individual are connected with a grey line and dots are coloured based on the age category (yellow: juveniles, green: adults). The red horizontal line indicates the age threshold between the categories (2 years). The number of combined amplicon sequence variants (cASV) per phylum for **B** the eukaryotic and **C** bacteria domain. **D,E** represent the relative abundance of each cASV (y-axis) for each sample (x-axis) for eukaryotes and bacteria, respectively. Samples are sorted by clustering of Bray-Curtis dissimilarity. Colours represent the different phyla, with fungi in yellow, parasites in blue, gray other eukaryotes and bacteria in red.

social rank differences, clan (same or different), seasonality (same season or different) and temporal distances (the distance in sample collection in days). Dry season was considered to apply from June to October and wet season from November to May[92]. We also accounted for possible sequencing batch effects (same vs different). We included two interactions: one between age and f-IgA and one between age and f-mucin, based on previous findings[65]. We performed a similar model restricted to juveniles (138 samples from 122 individuals) to assess the effect of sex. We applied Bayesian generalised linear multilevel models using the Markov chain Monte Carlo algorithm No-U-Turn Sampler (NUTS)[93] implemented in Stan probabilistic programming language through the brms R package v. 2.19.0[94]. We used a multi-membership random-effects framework that accounts for individuals in each pairwise comparison (e.g., Individual A, Individual B). All predictors were scaled to values ranging from 0 to 1 to allow comparison of the standardised estimates of the predictors. We used four Markov chains, with 4 chains, 3000 iterations, and 1000 burn-in iterations (warmup) to calibrate the sampler, and default uninformative priors. We visually inspected convergence and assessed the relevance of each predictor by analysing R-hat values and the 95% credible intervals (95% CI). R-hat values provide information about chain convergence - when below 1.01, they were accepted as indicators of good convergence. A parameter was considered "significant" when the 95% CI did not include zero. To verify the robustness of our results, we repeated all models using both occurrence, i.e. presence/absence (Jaccard distances) and abundance-based ß-diversity measures (Aitchison distances and Bray-Curtis distances), see Supplementary Table 1 - section Supplementary tables, supplements file. The results are largely congruent as indicated by the results in Supplementary Table 1, so we decided to report our results based on the Bray-Curtis distances.

We applied a similar model to each microbiome component similarity (bacteria, parasites, and fungi), with the same predictors as in the previous model in addition to the similarities of the other microbiome components.

We applied a complementary method to further investigate associations between immune measures (f-IgA and f-mucin), host characteristics (hyena ID, age at sampling, genetic mother), social and environmental factors (clan membership, social rank, season of sampling), the overall microbiome composition and possible sequencing batch effects. We implemented random forest regressions, a tree-based ensemble machine learning in the R package caret v. 6.0-94[95], and implemented the ranger method in the R package ranger v. 0.15.1[96]. The dataset was divided into training (80% of the samples) and test (the remaining 20%) sets using the function "createDataPartition", that used stratified sampling to create the splits, ensuring a representative distribution of the respective immune measure between the sets. Training and tuning were performed with the function "trainControl" using 10-fold cross-validation which was repeated 10 times. The final values used for the models were splitrule=variance and min.node.size=10 and mtry = 55 for the one predicting f-IgA, 266 for the f-mucin. We then evaluated the predictions by using the resulting model to predict each immune measure in the test dataset and compare them with the observed values by calculating a rank-based Spearman correlation. The importance of each variable was assessed based on the command argument importance = "permutation". This argument provides a variable's influence on the model's predictive performance. We applied partial dependence plots using the R package pdp v. 0.8.1[97] to evaluate the marginal effects of the top 20 most predictive variables for the f-IgA and f-mucin predictions and visualise the direction and linearity of these associations. As many taxa were not classified at phylum or even domain levels, we attempted to minimise uncertainty arising from this by inspecting the taxonomy of these taxa through searching for the most similar nucleotide sequences using NCBI BLAST searches[98] and inspecting the top hits and corresponding sample source. We have adjusted the taxonomic annotation accordingly as shown in Supplementary Data 2.

Co-abundance network analysis was used to infer potentially ecologically relevant taxa associations. Co-abundance networks were created with SParse InversE Covariance estimation for Ecological Association Inference (SPIEC-EASI), with the R package SpiecEasi[99] using the "mb" neighbourhood selection method. The optimal lambda coefficient for the network was

0.055. Network visualisation and topological evaluations were done using R package igraph v. 1.3.1[87]. We also calculated measures of centrality (Kleinberg´s hub, degree, closeness and betweenness) that help in positioning the individual contributions of each cASV in the network structure. These measures aim to identify potentially important taxa for the ecosystem structure and functioning, such as keystone species and lever species - species that shift the community structure[100].

## Reporting summary
Further information on research design is available in the Nature Portfolio Reporting Summary linked to this article.

## Results
### Intestinal communities of free-ranging hyenas
We profiled the intestinal community of 158 free-ranging hyenas from three clans in the Serengeti National Park, sampled from 2004 to 2018. Thirty-seven individuals were sampled several times throughout their lives (Fig. 1A - for more details on sampling design Supplementary Fig. 3 and for non parametric multidimensional scaling Supplementary Fig. 4). The median age of the animals at the time of sampling was 180 days, ranging from 52 days (1.7 months) to 5736 days (15.7 years). All adults were female (61 samples from 58 individuals), juveniles were of both sexes (138 samples from 41 males and 81 females).

The overall intestinal community was composed of 1597 cASVs deduced from 4706 ASVs across 35 different amplicons. 1183 cASV originated from eukarya, 407 cASVs originated from bacteria, including the phyla Firmicutes, Bacteroidota, Campylobacterota, Cyanobacteria, Proteobacteria, Planctomycetota, Fusobacteriota, Actinobacteriota, Deferribacterota, Spirochaetota, Desulfobacterota and unclassified (unknown) bacteria Fig. 1B–E). We identified 29 cASVs as known eukaryotic parasites of spotted hyenas annotated as Rhabditida (*Ancylostoma*) (7 cASVs), *Sarcocystis* (5 cASVs), Spirurida (3 cASVs), *Cystoisospora* (4 cASVs), *Cryptosporidium* (4 cASVs), Ascaridida (1 cASV), Diphyllobothriidea (2 cASV), Cyclophyllidea (3 cASVs). All but 5 samples had at least one parasite (cASV). Within the eukaryote, we also identified 690 cASVs as fungi, including the phyla Mucoromycota, Ascomycota, Basidiomycota, Blastocladiomycota, Chytridiomycota, Neocallimastigomycota. In addition, 7 cASVs were classified as unknown domain.

The intra-individual microbial difference was low or nil for the overall intestinal microbiome (dICC = 0.095, SE = 0.033), bacteria (dICC=0.060, SE = 0.050) and fungi (dICC = 0.022, SE = 0.021), but high for parasite composition (dICC = 0.123, SE = 0.085).

### Host immune measures are strongly associated with the intestinal microbiome
We evaluated the effect of various host and environmental factors on the overall microbiome membership (Jaccard distances) and structure (Aitchison and Bray-Curtis distances). We observed consistent effects for all ß-diversity metrics (Supplementary Table 1), we, hereafter, only report the results on community structure based on Bray-Curtis distances. Both mucus immunity and age related host factors exert the strongest impact on the overall intestinal microbiome (Supplementary Table 2, Fig. 2). Microbiome similarity decreased with increasing age differences (posterior mean −0.149, CI = −0.160 to −0.138) and increasing distances of f-IgA (posterior mean −0.050, CI = −0.062 to −0.038) and f-mucin (posterior mean −0.111, CI = −0.123 to −0.100) measures. Individuals with larger age differences and larger differences of immune measure levels tend to have a less similar microbiome than individuals closer in age and with more similar immune measures.

We further investigated the associations between immune measures and parasite, eukaryote, and fungi compositions. The bacteria and parasite similarities decreased with increasing f-IgA distances ([bacteria] posterior mean −0.100, CI = −0.116 to −0.083; [parasite] posterior estimate −0.062, CI = −0.087 to −0.037) (Supplementary Table 2, Fig. 3A). Fungi similarities did not substantially change with f-IgA distances (posterior mean=0.003,

**Fig. 2 | Host mucosal immunity and age are the major determinants of hyena gut microbiome composition.** The compositional differences (ß-diversity) of all taxa detected (overall microbiome, 1597 cASVs) was analysed between samples using a Bayesian regression multilevel model with Bray-Curtis similarity as the independent variable, based on pairwise distances among samples (n = 199 samples from 158 individuals). Among compared pairs, host effects were the distances between the levels of each immune measure (f-Iga, f-mucin), age differences, an interaction between age differences and each immune measure distances, and genetic mother's ID (same mother vs different mother). The social environment was investigated as the differences in the standardised social rank among each pair, ecological environment was included as the sampling temporal distances, seasonality (same vs different season) and clan membership (same vs different clan). Shown are the densities and boxplots of the posterior effect sizes. Individual ID of each pairwise comparison was included as multi membership random effect and sequencing batch effects was included as fixed effect in the model.

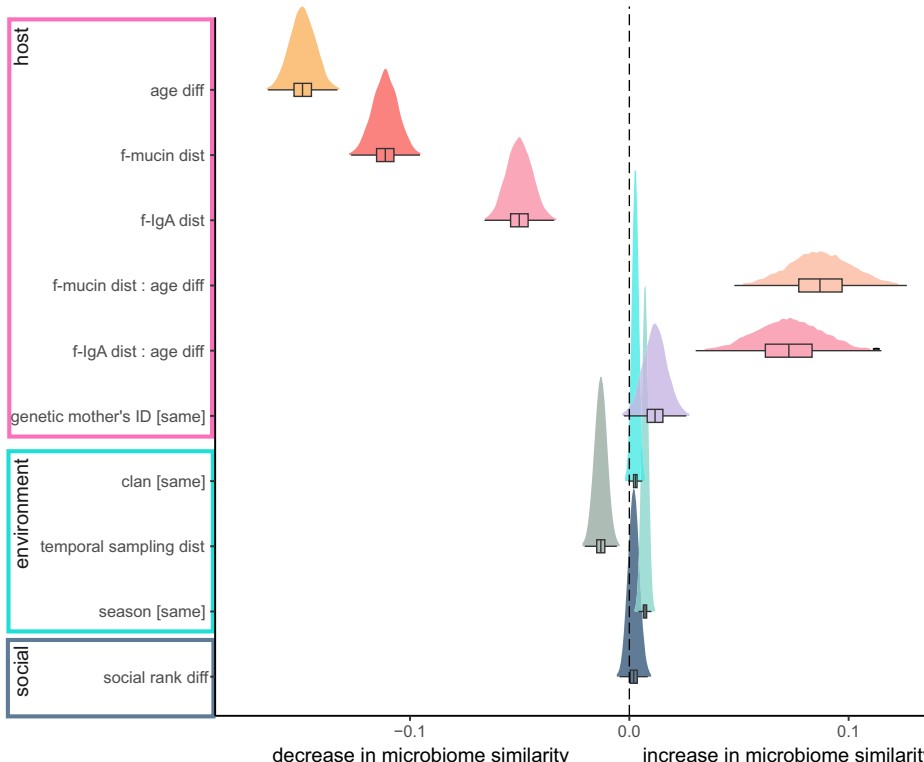

CI = −0.009 to 0.014). The bacteria, parasite and fungi similarities decreased with increasing f-mucin distances ([bacteria] posterior estimate −0.176, CI = −0.193 to −0.159; [parasite] posterior mean −0.078, CI = −0.104 to −0.053; [fungi] posterior mean −0.028, CI = −0.040 to −0.017) (Fig. 3B).

The effects of f-IgA on the parasite and also on the overall microbiome composition were modulated by age ([parasite] posterior estimate 0.142, CI = 0.077 to 0.211, [overall] posterior mean 0.073, CI = 0.041 to 0.105) (Supplementary Table 2). Similarly, the effect of f-mucin on parasite, fungi, and the overall microbiome similarities were also modulated by age ([fungi] posterior mean 0.046, CI = 0.017 to 0.076; [parasite] posterior mean 0.145, CI = 0.081 to 0.210, [overall] posterior mean 0.087, CI = 0.056 to 0.116) (Supplementary Table 2). This modulation means that, although microbiome similarity decreases with f-IgA or f-mucin distances, the rate of change (slope) is faster when individuals are more close in age than when individuals have larger age differences (Fig. 3C, D). In other words, the effect of immune measures are more pronounced when individuals have smaller age than larger age differences.

Age strongly affected the overall microbiome composition (Fig. 2) but also the bacteria, parasite, and fungi compositions. The microbiome similarity was higher when age differences were smaller for all groups assessed ([overall] posterior mean −0.149, CI = −0.160 to −0.138 [bacteria] posterior mean −0.161, CI = −0.178 to −0.144; [parasite] posterior mean −0.112, CI = −0.138 to −0.087) except for fungi similarity, that decreased with smaller age differences ([fungi] posterior mean 0.015, CI = 0.004 to 0.028) (Supplementary Table 2).

Sharing the same genetic mother had a positive effect on the overall microbiome similarity (posterior mean 0.012, CI = 0.001 to 0.022), and, within juveniles, we find no effect of sex on the overall microbiome composition and also on the bacteria, parasites, and fungi (Supplementary Table 3).

**Environmental factors affect the gut microbiome composition**
Environmental effects also shape the overall microbiome composition, but also bacteria and fungi compositions. Temporal distance of sample collection, i.e. the amount of time elapsed, in days, between the collection of two samples, decreased the overall microbiome similarity (posterior mean

−0.013, CI = −0.018 to −0.008) and the fungi similarity (posterior mean −0.024, CI = −0.029 to −0.019). Samples collected in the same season had a more similar overall microbiome (posterior mean 0.007, CI = 0.005 to 0.009), bacteria (posterior mean 0.107, CI = 0.004 to 0.010) and fungi (posterior mean 0.012, CI = 0.009 to 0.014). Individuals from the same clan had a more similar overall microbiome (posterior mean 0.003, CI = 0.0004 to 0.005) (Supplementary Table 2, Fig. 2).

**Stronger associations of specific taxa and immune measures**
To further investigate the association between the intestinal microbiome and mucosal immunity, we applied a random forest regression to the relative abundances of all 1597 cASV identified to predict f-IgA and f-mucin. The 199 samples were divided into training (80%) and testing (20%) sets. Predicted and observed values correlated moderately for f-IgA ($R^2 = 0.339$, Spearman's rho = 0.607, $p < 0.001$, n = 39) and f-mucin ($R^2 = 0.338$; Spearman's rho = 0.574, $p < 0.001$, n = 39). We identified the top 20 most predictive variables according to the random forest importance score for f-IgA (Fig. 4A) and f-mucin levels (Fig. 4B). Apart from age, the most predictive variables were taxa abundances. The vast majority of these are similar to other mammalian gut taxa, see Supplementary Data 2. These taxa are positively or negatively associated with f-IgA and/or f-mucin (Fig. 4C) and many with non-linear associations with the corresponding immune measure (Supplementary Figs. 5 and 6).

Among the highest predictive taxa of f-IgA levels were age, 16 bacteria cASVs, 2 parasite cASVs (both *Ancylostoma*), and 1 fungi cASV (*Pleosporales*) (Fig. 4A). For f-mucin levels, the highest predictive taxa were 18 bacteria cASVs and one fungi cASV (*Pleosporales*) (Fig. 4B). Out of these, the same 6 cASV predicted both f-IgA and f-mucin levels, in similar directions: cASV 8 and 801 (uncultured bacteria), cASV 674 *Bacteroides*, cASV 739 *Collinsella*, cASV 39 *Dorea* and cASV 392 *Erysipelotrichacea* (Fig. 4C).

**Strong associations among gut taxa**
We investigated the effects of inter-taxa interactions by, firstly, investigating the effect of group similarity (bacteria, parasites, and fungi) on each other. All groups were associated with each other. Bacteria similarity increased

**Fig. 3 | Mucosal immunity exerts a larger effect on the intestinal bacterial community composition than on parasite and fungi communities.** The compositional differences (ß-diversity) of all taxa detected (overall microbiome, 1597 cASVs, green), bacteria (407 cASVs, red), parasites (29 cASVs, blue) and fungi (690 cASVs, yellow) were analysed between samples using a Bayesian regression model with Bray-Curtis similarity as independent variables. Shown are the posterior distributions for the effects of f-IgA (**A**) and f-mucin distances (**B**). Dots represent mean effect sizes, and estimates and bars represent the corresponding 95% credible interval (CI) on intestinal community composition similarity. Note that the ranges on the x-axis differ between the plots, reflecting different effect sizes. Predicted values based on the Bayesian regression models for the association between the overall microbiome similarity and (**C**) f-IgA and (**D**) f-mucin distances, modulated by age distances. The predicted similarity of the intestinal microbiome is displayed between sample pairs with mean age distances ± 1 standard deviation (SD) (mean age distance = 0.16 ± − 0.18). The lines indicate the mean predicted immune measures and the shaded areas represent the 50% credible intervals (CI). For instance, for both immune measures, the highest overall microbiome similarity is between animals with similar levels of f-IgA and f-mucin and similar age.

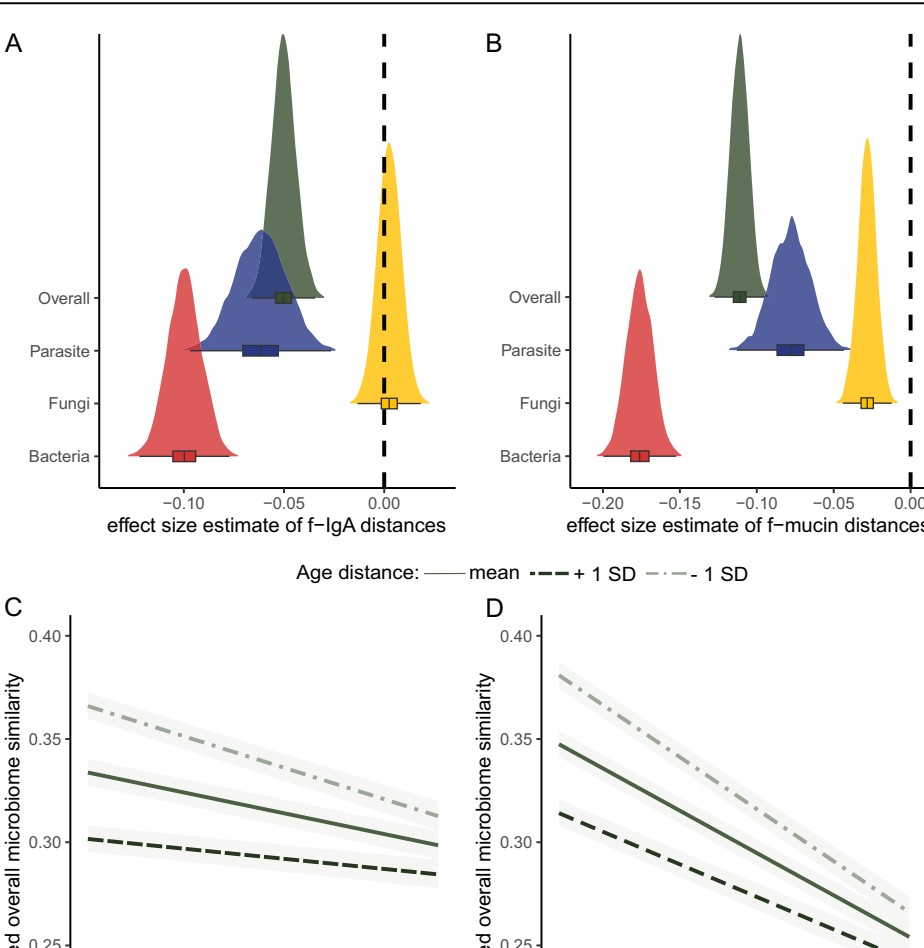

with parasite (posterior mean 0.107, CI = 0.097 to 0.11) and fungi similarities (posterior mean 0.153, CI = 0.133 to 0.173). Fungi similarities increased with bacteria (posterior mean 0.071, CI = 0.062 to 0.081) and parasite similarities (posterior mean 0.028, CI = 0.022 to 0.035). Parasite similarity increased with bacteria (posterior mean 0.240, CI = 0.219 − 0.260) and fungi (posterior mean 0.135, CI = 0.104 to 0.165) similarities (Supplementary Table 2).

We further explored inter-taxa interactions with a co-abundance network of all cASVs. The resulting network contained 1597 nodes and 24614 edges (Fig. 4D). The network revealed a highly connected community. The most predictive taxa of either f-IgA and f-mucin had higher Kleinberg's hub centrality scores (mean= 0.56, standard deviation (sd) =0.18) than other taxa (mean = 0.24, sd = 0.17, F = 92.68, $p < 0.0001$), higher degree centrality (mean=58.55, sd = 16.00 vs mean = 30.31, sd = 17.31, F = 75.8, $p < 0.0001$), higher closeness centrality (mean = 0.00062, sd = 0.00002 vs mean 0.00058, sd=0.00002, F = 24.75, $p < 0.0001$) and no differences in betweenness centrality (mean = 1630, sd = 881 vs 1561, sd = 1576).

## Discussion

We explored the associations between two important and broad-acting measures of intestinal mucosal immunity, IgA and mucin, and the gut microbiome, while accounting for host characteristics (i.e. age, ID, genetic mother), social rank, and environmental factors (i.e. sampling date, seasonality and clan membership) in Serengeti hyenas. Both IgA and mucin

were strongly associated with the composition of the intestinal community, and these associations varied in strength within different components of the gut microbiome, being stronger within bacteria, intermediate within parasites and weaker within fungi communities. The most important taxa predicting both immune measures were bacteria, the parasite *Ancylostoma,* and Pleosporales fungi, including the genus *Phaeosphaeria*. Our results highlight the complex relationships between mucosal immunity and the gut microbiome, considering both bacteria and eukaryotes, in a wild mammal population.

Bacteria and eukaryotic parasite compositions were strongly associated with both f-IgA and f-mucin, and fungi composition was moderately associated with f-mucin. Bacteria outnumber eukaryotes in mammalian intestines[101–103] and in the metabolic products that are currently known to interact with the immune system[104,105]. Known eukaryotic parasites in this study comprise fewer taxa than bacteria and fungi but were still strongly associated with both immune measures, highlighting the strong relationship between parasites and host immunity. This is expected, because parasites can tamper with immune responses to their own advantage, e.g. by secreting enzymes that degrade the mucin layer[106]. The interactions between fungi and mucosal immunity are still poorly understood, but recent studies have revealed potentially relevant associations[23,107].

We explored which taxa showed the strongest associations with immune measures. Many of the identified taxa are well-known intestinal symbionts in humans and mouse models, associated with either health

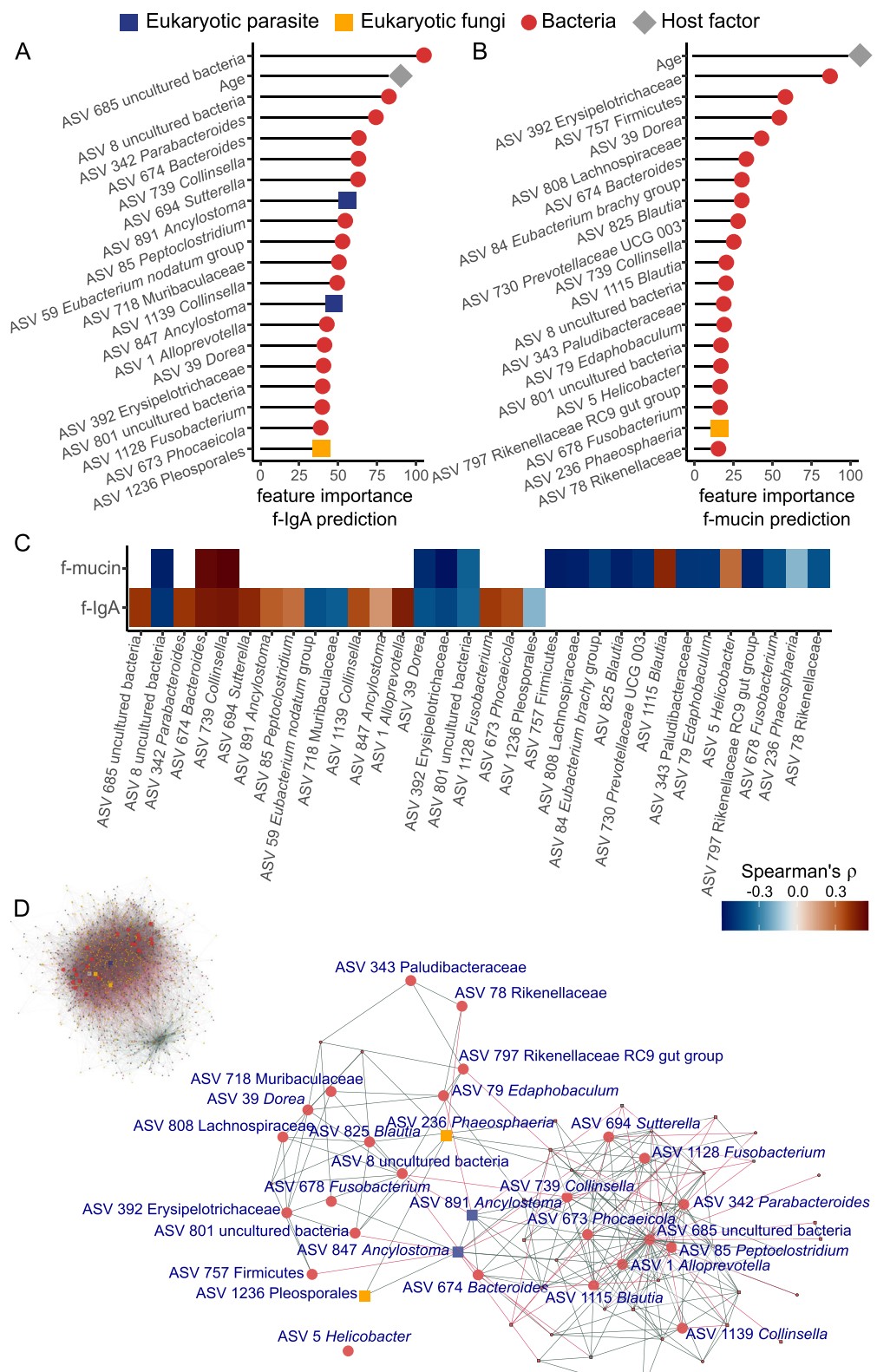

**Fig. 4 | Age and specific intestinal bacterial, parasitic and fungi components predict levels of faecal markers of mucosal immunity (IgA and mucin) in spotted hyenas.** Random forest regressions were implemented with 1597 combined amplicon sequence variants (cASVs) from 160 faecal samples (training set) and evaluated with an independent dataset of 39 samples (test set), to predict IgA and mucin levels. The top 20 most important cASVs for model prediction of (**A**) f-IgA and (**B**) f-mucin levels. Red represents taxa from the bacterial domain, blue parasites, yellow fungi, and grey host. **C** A heatmap illustrating the directionality of the relationships (negative in blue, positive in red) between the two immune measures and the top 20 most important cASVs for predicting f-IgA and f-mucin. **D** Co-occurrence network of 1597 taxa across 199 samples from 158 spotted hyenas. Nodes represent a cASV and edges the associations between a pair of cASVs. In more detail is a section of the co-occurrence network of the most important taxa for predicting both f-IgA and f-mucin. In both networks the green and red edges (lines) represent positive and negative associations, respectively.

benefits or detrimental effects. Some of these bacterial taxa, including *Bacteroides, Alloprevotella, Peptoclostridium*, and *Fusobacterium*, were previously identified as 'core gut symbionts', i.e. present in 85% of samples, from a different population of spotted hyenas[64]. It is conceivable that these taxa provide relevant functions to hyenas. The *Bacteroides* genus is a common symbiont of humans[108] and other mammals[109–111] and its members have been associated with beneficial[112] and negative effects in their host[113]. Although the functional role in intestinal health is still largely unknown, *Alloprevotella* was suggested to have anti-inflammatory properties in humans[114], was reduced after a stressful intervention in pigs (*Sus domesticus*)[115], and was associated with the presence of helminths in wild chimpanzees (*Pan troglodytes*)[116]. Taxa of the *Fusobacterium* genus are associated with disease states in humans and in its translational models. *F. nucleatum* causes inflammation by upregulating the activity of IgA in laboratory mice[117,118] and worsens colorectal cancer in humans[119,120]. Members of the genus *Peptoclostridium* can help maintain the intestinal barrier, among other functions[121], but can also be life-threatening to humans and other animals[122].

We found the parasite *Ancylostoma* to be positively associated with f-IgA levels. *Ancylostoma* is a blood-feeding nematode that causes extensive inflammation and intestinal bleeding in several host species[123–125]. Our findings are consistent with those of a previous study in the same population, in which *Ancylostoma* egg load was positively associated with f-IgA[65]. We interpret this result as the potential upregulation of IgA by the host in an unsuccessful attempt to eliminate or contain the scale of the infection by adult *Ancylostoma* parasites from the gut.

The composition of parasite, fungi and bacteria components were dependent on each other. Samples with more similar parasite composition also had more similar bacteria and fungi compositions, and vice-versa. When exploring inter-taxa correlations with co-occurrence network analysis, a highly connected intestinal community emerged. The taxa predicting both f-IgA and f-mucin have higher centrality (hub, degree and closeness) than other taxa, indicating the potential of these taxa to have important ecological roles and a regulatory role of the mucosal immune system. Previous studies in wild non-human primates have also documented associations between parasites and bacteria, which are shaped by host-related and ecological factors[126–128]. The role and significance of fungi within the intestinal community remains to be elucidated given the scarcity of studies focusing on this taxonomic group within wildlife guts, but see refs. 40,126.

As expected from previous studies on hyenas[35,65] and other wild animals (e.g. wild primates[129]), we found that age is a strong predictor of microbiome similarity. Our results suggest that the strength of the associations between bacteria, parasites, and fungi on the one hand and mucosal immune measures on the other were modulated by host age. These effects were complex and inconsistent between measures of mucosal immunity and among the different components of the gut microbiome. The mammalian gastrointestinal tract is colonised at birth by pioneer microbes acquired from mothers and the environment[130,131], and the gut microbiome undergoes a process of microbial succession[132]. This process is closely linked to the maturation of the immune system, as the immune system requires microbial interactions early in life for proper development and maturation, and in turn shapes the microbiome composition[9,133]. Our results are consistent with complex and context-specific interactions between the immune response and the host throughout life, particularly during juvenile stage and the transition to adulthood.

It is difficult to comment on the health effects of a particular gut microbiome community or even on those of individual taxon and associations, other than of known parasites such as *Ancylostoma*[65]. Thus, we are cautious when interpreting the functional roles of individual taxa associated with mucosal immunity in this study. Different species, strains, and immunogenic potential within these taxa could lead to different short and longer-term outcomes for individuals in wild populations, for which knowledge is still limited. Host-microbial interactions are context-dependent and are mostly studied in humans and laboratory translational models. In future studies, assessments of the effects of particular microbial

communities on individual health (as measured by body condition) or fitness proxies (e.g. survival), may help determine whether specific taxa are beneficial or not to hosts, and thus involved in shaping host evolutionary fitness. Because we measured broad-acting measures of mucosal immunity, we cannot infer specific mechanisms of interactions between the different members of the gut microbiome and the host. In future, measuring antigen-specific IgA isoforms and mucin types together with assessments of intestinal communities may shed light on the finer details of the regulation of individual taxa in the biomes. In the broad absence of reagents for wildlife antibody detection[134], IgA-seq, the enrichment of sequencing of IgA-coated taxa[135,136] could provide unprecedented insights into taxon-specific immune control at intestinal barriers. Investigating the link between host fitness, the microbiome and its metabolites could reveal evolutionary adaptations that promote or hamper host health and performance.

Natural populations harbour a hidden and mostly unknown diversity within their guts, and their immune systems must regulate such communities by maintaining mutualists and commensals and reducing detrimental parasitic interactions. We identified broad and general associations between immune measures and the different components of the gut microbiome and pinpointed the taxa driving these associations. These findings indicate the important role that the immune system plays in both the defence and regulation of the microbiome, and we propose that the identified taxa are closely associated with and involved in the cross-talk within the gut of wild populations of hyenas - a potential product of co-adaptation. The next step is to further investigate the genetic diversity and functional profiling of gut microbiomes in natural populations to uncover evolutionary aspects of such potential co-adaptations. We thus encourage others to approach wildlife microbiome research in a holistic manner and incorporate local mucosal measures of the immune system to improve our understanding of the complex and dynamic interactions between hosts and their gut microbiomes.

## Data availability
All data used in statistical analyses and figures is openly available from Zenodo Digital Repository (https://doi.org/10.5281/zenodo.15283097)[137]. In addition, all raw sequencing data can be accessed through the BioProject PRJNA1134446 in the NCBI SRA.

## Code availability
All code used for analyses is available through the Zenodo Digital Repository (https://doi.org/10.5281/zenodo.15283097)[137].

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

## Acknowledgements

For fieldwork, we were granted research permits from the Tanzania Commission for Science and Technology (COSTECH) and permission from the Tanzanian National Parks Authority (TANAPA) and Tanzanian Wildlife Research Institute (TAWIRI). Fieldwork was supported by the Commission for Science and Technology of Tanzania (COSTECH), the Tanzania Wildlife Research Institute (TAWIRI), and Tanzania National Parks (TANAPA). For laboratory support at the Leibniz Institute for Zoo and Wildlife Research (IZW), we thank D. Thierer, K. Pohle, F. Webster and C. Bost. We thank A. Francis, T. Shabani, M. Andris, N. Boyer, T. Golla, K. Goller, N. Gusset-Burgener, B. Kostka, M. Lindson, D. Thierer, A. Türk and K. Wilhelm for field and technical assistance. Hyena icons were designed by Sonja Metzger. We thank the Leibniz-Association for funding the EpiRank project (grant SAW-2018-IZW-3-EpiRank). We thank the Deutsche Forschungsgemeinschaft DFG (grants EA 5/3-1, KR 4266/2-1, DFG-Grako 1121, 2046), the Leibniz Institute for Zoo and Wildlife Research, the Fritz-Thyssen-Stiftung, the Stifterverband der deutschen Wissenschaft and the Max-Planck-Gesellschaft and Research Institute of Wildlife Ecology, University of

Veterinary Medicine Vienna (SCMF) for financial support of the project. This work was also supported by the Deutsche Forschungsgemeinschaft (DFG) (Grant Number: 285969495/HE 7320/2–1), the German Academic Exchange Service (DAAD) (VHJD scholarship holder during PhD studies) and the Research Training Group 2046 "Parasite Infections: From Experimental Models to Natural Systems" (RTG-GRK2046: SPVS, MMV and VHJD as PhD students and E.H., H.H., M.L.E., G.A.C. and S.B. as Senior Researchers).

## Author contributions

E.H., G.A.C., S.B., A.W., M.L.E. and H.H. conceptualised the original study and acquired funding. S.C.M.F., S.P.V.S., E.H. and S.B. designed the analyses and computational framework. S.C.M.F., S.M., M.L.E., H.H. and S.B. conducted fieldwork. S.C.M.F. and M.M.V. performed the laboratory work for immune measures. V.H.J.D. and S.K. performed laboratory work for microbiome analyses. S.C.M.F. and E.H. analysed the data. S.C.M.F. and S.P.V.S. wrote the manuscript with contribution and feedback from S.B. and E.H. All authors contributed significantly to editing the manuscript.

## Funding

## Competing interests

The authors declare no competing interests.

## Additional information

[1]Department of Ecological Dynamics, Leibniz Institute for Zoo and Wildlife Research (IZW), Alfred-Kowalke-Strasse 17, 10315 Berlin, Germany. [2]Department of Wildlife Diseases, IZW, Alfred-Kowalke-Strasse 17, 10315 Berlin, Germany. [3]Institute for Biology, Department of Molecular Parasitology, Humboldt University Berlin (HU), Philippstr. 13, Haus 14, 10115 Berlin, Germany. [4]Research Group Ecology and Evolution of Molecular Parasite-Host Interactions, IZW, Alfred-Kowalke-Straße 17, 10315 Berlin, Germany. [5]Department of Evolutionary Genetics, IZW, Alfred-Kowalke-Straße 17, 10315 Berlin, Germany. [6]German Centre for Integrative Biodiversity Research (iDiv), Halle-Jena-Leipzig, Puschstrasse 4, 04103 Leipzig, Germany. [7]Department of Biology, Chemistry, Pharmacy, Freie Universität Berlin, Arnimallee 22, 14195 Berlin, Germany. [8]Department of Veterinary Medicine, Freie Universität Berlin, Oertzenweg 19b, 14163 Berlin, Germany. [9]Leibniz Institute for Zoo and Wildlife Research (IZW), Alfred-Kowalke-Strasse 17, 10315 Berlin, Germany. [10]Research Institute of Wildlife Ecology, Department of Interdisciplinary Life Sciences, University of Veterinary Medicine Vienna, Savoyenstrasse 1, 1160 Vienna, Austria. [11]Present address: Max-Delbrück-Center for Molecular Medicine in the Helmholtz Association (MDC), Robert-Rössle-Str. 10, 13125 Berlin, Germany. ✉e-mail: soares@izw-berlin.de; benhaiem@izw-berlin.de; susana.ferreira@vetmeduni.ac.at

