## [Transparent Peer Review file · Communications Biology]

Mucosal immune responses and intestinal microbiome associations in wild spotted hyenas (*Crocuta crocuta*)

Corresponding Author: Ms Susana Patrícia Veloso Soares

Version 0:

Reviewer comments:

Reviewer #1

(Remarks to the Author)

Journal: Communications Biology

Manuscript #: COMMSBIO-24-4557-T

Title: Spotted hyena gut cross-talks: Symbionts modulate mucosal immunity

Current revision: v0

Due: Sept 4th, 2024

Manuscript summary: In this study, the authors investigated immune measures (IgA and mucin) and used a multi-amplicon approach to profile bacterial, fungal and eukaryotic parasitic communities in 199 fecal samples collected from 158 wild hyenas over a ~14 year period. This work follows up previous studies that demonstrated individualized microbiomes in this population (influenced by factors such as age and prey availability), and a positive correlation between fecal IgA and mucin concentrations and *Ancylostoma* egg burden (also correlated with age and host survival). The goal of the current manuscript was to determine the relationship between immune measures and gut microbial community composition. The hypothesis was that host-associated microbial community composition was strongly associated with fecal IgA and mucin levels. The authors used probabilistic models and machine learning in their analysis. Their main findings were: 1) immune measures predicted microbiome similarity in an age-dependent manner; 2) immune measures were more strongly correlated with the bacteriome than parasites and even more so than fungi and 3) microbiome composition could be used to predict immune levels. Specific taxa driving this association were identified, including *Ancylostoma*.

Overall impression of the work: This study leveraged a dataset that was impressive both in terms of sample size and study duration. Given the recent research coming from the same group of researchers that has shown this hyena population has individualized fecal microbiotas and fecal IgA/mucins levels/parasite burdens associated with survivorship, it seemed fitting that this study bridge these factors by linking IgA/mucin to microbial diversity. That said, I found it hard to keep track of which taxa were being targeted, especially because certain results highlighted taxa that I had assumed were off-target (vertebrate DNA and organisms unclassified at the level of domain/phyla). Admittedly, I have not worked with multi-amplicon microfluidic PCR myself, but without a clear description of data normalization and consistent data subsetting, I was not convinced that the relative abundance of certain microbial groups was directly comparable. I also felt that the bulk of the figures revolved around the Bayesian probabilistic model and pairwise comparisons of both diversity measures and covariate measures, which are not intuitive for the reader. I believe that the incorporation of more straightforward visualizations along with such analyses will be more effective. Overall, I think this manuscript has great potential but will require some work to clarify the methods and better visualize the results.

COMMENTS TO THE AUTHORS:

Major comments:

Data reproducibility & confusion about methods: The following issues made it difficult for me to fully understand and evaluate what was done in this analysis.

Code: Thank you for providing code, but it looks like rFit_IgA.rds is missing from the tmp folder so I was unable to reproduce the latter part of the analysis. I'm also unable to link the sequences to the ASV IDs used in the manuscript when starting with the fPMS.rds phyloseq object. (Minor note- for the next submission, please organize the code according to the figure numbers so chunks are easier to locate. For example, the code refers to "Fig5" but there is no Fig 5.)

Primer selection: I'm confused about which taxa were targeted in this study. The primer table is missing some information and some primers aren't mentioned in the manuscript (i.e., CytB, HSP70, tRNA and COI). Were all these primers used in this study? Was qPCR done? Was host DNA targeted in addition to microbial? The manuscript divides the data into 3 groups: bacteria, fungi & parasites, so I would tidy the primer table and organize it as such. In addition to the broad "Target" column, add detail what was targeted by specific primers at a finer level (e.g., protozoa/helminths/fungi within Eukaryota). Add citations to back up primer choices. This should be provided as a supplemental table/file and I would consider adding a simplified table to the main manuscript. Below are some related comments regarding the associated text that may clear up confusion:

- o Lines 187-191 are redundant. Remove the first sentence and clarify the targets of each (e.g., "the UNITE database for fungal ITS...").
- o Line 205 refers to "four groups of cASVs" but only three are listed. Plus, are they all really "cASVs"? For example, only one primer pair was used to target 28S, so wouldn't these sequences just be "ASVs"? If both cASVs and ASVs were used this should also be specified in the primer table.
- o Lines 205-211: Just list the three broad categories (bacteria, fungi & parasites). Why are specific phyla are listed here? If these were the targets, include that information in the primer table. If these are lists of the successfully sequenced phyla, this information should be in the results.

Non-microbial reads and unclassified taxa: I'm not sure why the authors decided to retain vertebrate DNA in the analysis since the focus was on the microbiome. They also retained unclassified ASVs. Since these are typically filtered out early in the analysis, I suspect removing these ASVs may significantly affect the results.

Off-target amplification: Was the hyena host DNA specifically targeted here? Or was this host contamination as is common with certain primers? Most microbiome studies aim to avoid amplifying host DNA so that a larger proportion of the reads are microbial targets. Some even use PCR blockers to avoid host contamination. If the host DNA was of interest, wouldn't quantification be more important (i.e., qPCR, ddPCR, WGS)? If the goal of using host DNA was to examine inflammation, wouldn't immune assays or even stool blood content be more appropriate? Furthermore, there I don't see a "Crocuta" designation in the tax table and the only cASV from Vertebrata. This BLASTs to many mammalian species (e.g., 1st hit is an Etruscan shrew). How is it known that this is hyena DNA and not prey DNA? What is more confusing is that at line 486 in the discussion the authors state "our results indicate that prey DNA is negatively correlated with host DNA". I cannot find any analysis to support this claim. How could this comparison be made with only one cASV corresponding to Vertebrata? Several cASVs correspond to arthropods, but wild hyenas must be eating vertebrates as well.

Unclassified taxa: What was the rationalization behind keep taxa with unclassified kingdoms and domains? Typically, taxa that aren't assigned to the level of phyla are removed prior to the data analysis since they may be spurious. This is especially curious considering many of these taxa were identified in the machine learning model. I appreciate that the BLAST table was provided, but the fact that so many hits were to unclassified bacteria or even "unclassified organism" makes me think they should have been filtered.

Merging ASVs: Line 201 refers to the Ferreira et al., 2023 manuscript for the detailed methods of merging ASVs. However, it appears that in that study the tree was trimmed down to just Eimeria. The current analysis seems more complicated since multiple primers were used on multiple targets. Perhaps another reference is missing from the claim in line 202 "is now extended to all annotated genera" that contains such a workflow? If not, I recommend rephrasing that to something like "and is extended to multiple microbial targets in the current study" and expanding on the workflow description (e.g., including supplemental trees, perhaps before and after merging into cASVs).

Normalization: How was this data normalized? The all-in-one approach of using the Fluidigm Assay has obvious benefits, but one would assume that different primers have different ideal conditions, and without control over individual reactions, bias might be introduced. In Ferreira et al., 2023, qPCR was used. Was it used here as well? Even so, the previous study was only concerned with Eimeria, so how did the authors in the current study account for multiple kingdoms of life? For example, in Fig 2A,B the data from eukaryotes is lumped together, but given it spans 18S, 28S and ITS, I'm not sure that this is appropriate (see comments on Figure 2).

Data subsetting: From the code it seems that parasites and fungi were subset down to select taxa, while bacteria were not. What was the rationale? Isn't the goal to get a wholistic view of the microbial community? If the data was not normalized and then certain clades were subset even further, I'm not sure it is appropriate to compare across groups. For example, one main claim is that bacteria had a stronger association on immune measures than parasites and fungi, but perhaps that is because the entire bacteriome was considered versus just a subset. Line 301 says data was subset to "known parasites"- what about other microeukaryotes that might correlated with ill health? Also, how were there pathogens selected? Toxoplasma, a known hyena parasite, was left off this list but was detected in the study. What about microeukaryotes that may have beneficial roles (see: <https://www.frontiersin.org/journals/microbiology/articles/10.3389/fmicb.2023.1123513/full>)?

Sample metadata: It's unclear how many runs were performed or whether data was analyzed for batch effect. The metadata

should reflect which run each sample came from as well as the negative controls. DNA concentrations used in Decontam (are these from Qubit or qPCR?) should be included as well. Were technical replicates included (i.e., same sample, different run/reaction)? If so, these should be specified in the metadata (even if they weren't all used in the main analysis)

Confounding age & sex covariates: Why were only adult females sampled? The rationale here should be described. Was it not possible to sample adult males? From Ferreira et al., 2021: "Fecal concentrations of IgA, IgG, and mucin increased with Ancylostoma egg load and were higher in juveniles than in adults. Females had higher mucin concentrations than males". The fact that all adults in this study are female makes it more difficult to parse out the influence of sex, age, and immune measures on the microbiome.

FIGURES: In general, I felt that there was redundancy in the figures, while some important concepts were not visualized. Specifically, Table 1 conveys the same information as Figs 2E,F and Fig 3A-C. The goal of the paper is to show an association between immune measures and microbial diversity, yet we don't see plots of alpha and beta diversity measures over IgA/mucin levels aside from the pairwise comparisons (which can be hard to digest). In addition, although temporal distance between samples is significant for all microbial groups in Table 1, only age is further analyzed.

FIGURE 1:

- I don't find the visualizations of the pairwise distances in B,D and F that interesting, they could go in the supplemental. Instead, I'd rather see the histograms for the clans, social ranks and year of sampling from Fig S1 since these are the covariates used in the model in Table 1.
- I liked the hyena icons at first, but now feel like they are distracting. This is especially true for 1B since according to the dyad.data.rds there aren't any adult pairs at Age distance=0 (only juvenile-juvenile pairs). I think it's best to just note "more similar in age" or "similar f-IgA levels" with an arrow pointing towards "dissimilar ages" or something along those lines...
- o Legend: The abbreviation RU is defined, yet I do not see it used in the figure. I believe "immune measures levels" should be "immune measure levels".
- o 1A: This should be stretched out or divided into hyenas sampled once and those with repeat sampling because right now it is hard to see what is happening up top.
- o 1B: I'd label the x axis "Pairwise age distances"
- o 1D: Incorrectly labeled 1E in legend. I'd label the x axis "Pairwise f-IgA distances" or "dissimilarities"
- o 1E: Incorrectly labeled 1D in legend

TABLE 1: First off, I'd start with the relative abundance plots (Fig 2A-D) and then show how these communities are associated with immune measures. If using this table (instead of Figs 2E,F & Fig 3A-C), I'd clean it up a bit by making it more obvious which are significant (like the supplemental table) and adding labels on the side to delineate the immune measures, social/environmental covariates and interactions. Shouldn't a sex:mucin interaction be added given previous findings (see above)? Perhaps you could repeat the analysis on just juveniles to investigate sex.

- o TABLE 1 and Figs 2E,F & Fig 3A-C: How does one interpret these results in plain terms? Most tables like this list the covariates as predictors, not the pairwise distance between covariates. And is there directionality associated with +/- values? That should be clearly labeled. The figures are also confusing because the x-axis is labeled with the covariate distance, but I believe it should be "BRM Estimate size". For example: Fig 2E has the x-axis title "f-IgA distance" yet the values are all negative and the f-IgA distances shown in 1D are all positive.

FIGURE 2: Fig 2A-D feel like one figure about relative abundance, while E,F pertain more to Table 1.

- o 2A,B: Again, for the reasons discussed above I don't think it is appropriate to plot these microbial groups together. This is especially true in B where the relative abundance adds up to 100% (which is arbitrary). At least split these up into parasites and fungi.
- o 2E: These values (Overall, Parasite) don't match the table. Your code does provide the correct version, however.
- o 2E,F: If you keep these I'd make the x-axes consistent so the differences are more clear.

FIGURE 3:

- o 3A-C: I prefer table 1 since it is more obvious which are significant. If keeping these plots, I'd lighten the bars that cross 0 to highlight the significant groups
- o 3D,E: Can you clarify what was done here? Did you simply take the model from Table 1 and enter either AgeDistance=0 or 1 and then generate predictions along set intervals from f-IgA 0 to 1 (similar to dissimilar pairwise f-IgA comparisons?). This is hard to interpret without the number of samples and individuals used. Again, there are no adult-adult with AgeDistance=0 in the dyad.data, so 0 is only juveniles (both are shown in the legend). Also, there is only one pairwise comparison where AgeDistance=0 (sample P816_B8320 and M148_X5461). I'm not sure this is the best way to present the data. We already know from previous research that the microbiome of adults and young juveniles should look different and that IgA/mucin is associated with age in hyenas. I'd prefer a visualization of your data than model predictions. I think it would be more effective to stratify by age groups and look for associations within each. Perhaps just ordinate the microbial diversity for juveniles, see which axis separates the data by f-IgA and plot the axis loadings over f-IgA (then repeat for adults).

Longitudinal analyses: I found it odd that even though temporal distance (time) was found to be significantly influential for all microbial groups tested, this data wasn't visualized. Was seasonality explored at all? I'd expect it might affect parasite load. I'd love to see visualizations showing microbial diversity metrics (α diversity and beta diversity ordination loadings), the abundance of microbial groups, and immune measure levels over age AND over time.

- o How were longitudinal samples handled? Including multiple samples from the same animals could inflate the association between the microbiome and immune measures. These longitudinal samples could also be used to analyze within individual changes over time (especially the individuals with 3 samples) although this may be outside the scope of this

analysis.

FIGURE 4: See above for comments related to the host & unclassified DNA...

o 4A,C: I don't find these graphs very interesting aside from the R2 value, they could be supplemental. What is the shading? Are these credible or confidence intervals?

o 4B,D: These plots are more interesting but don't give information on directionality. I'd suggest that rather than sorting by importance, you provide a heatmap of samples sorted by f-IgA or mucin measurement levels and the relative abundance of these taxa

o 4E: I like the use of a network but would like to see it without host DNA. Also, please keep the colors chosen for your microbe groups consistent throughout the figures.

Minor comments:

- Title: "Symbiont" doesn't seem appropriate here, especially since the data selection for the eukaryotes was geared towards parasites. Also be careful of implying causation.

- Line 57: "Intestinal mucosa" often refers to the GI layer (i.e., the epithelium, lamina propria & muscularis). Did you mean the "intestinal mucosal epithelium" of the small intestines/colon (remember that distal GI tract regions have stratified epithelium). Keep in mind which regions fecal samples likely represent.

- Line 61: Recommend "can be associated" since perturbations can also lead to a new baseline community that is not necessarily unhealthy (e.g., a diet shift)

- Line 78: "inter-kingdom"?

- Line 83: Over-generalization. This sentence refers to "wild animals" in captivity- in the context of this study, readers may assume "captivity" means zoos. However, two referenced studies (27, 29) are on mice (I'd argue lab mice have unique husbandry) and the other sampled humans (28). The latter included published data on wild/captive mammals, yet wasn't designed to explore captivity. Ley found that host phylogeny/diet overshadowed other covariates. More recent studies of wild mammal bacteriomes suggest that certain host clades (e.g., Perissodactyls) are resistant to the captivity-induced changes (see doi: 10.3389/fevo.2021.785089 and DOI: 10.1093/icb/icx090). Suggest rewording and/or adjusting references.

- Line 87: Flow is a bit confusing. Line 80 gives a nice transition to wild mammals, but then 83 refers to mice/humans and then 87 is vague (yet the refs have transitioned to studies of wild animals). IMO, this paragraph should focus on the growing knowledge surrounding wild mammalian population microbiomes, the feasibility of non-invasive sampling and the implications for conservation or understanding microbiome-environmental interactions.

- Lines 106-111: Really nice job stating the hypothesis and tying in the need for mixed community microbiome analysis using refs to previous microbiome work and the *Anycyclostoma*/mortality example! I would try to flesh out the sentence about the previous microbiome, parasite and immune measure studies that were recently done (I believe by your lab) to make it clear what findings informed your current work and how this study helps complete the story.

- Lines 115-125: Impressive dataset and I appreciate the use of host genotyping. Is this the only sentence referring to the host genotyping methods? The reference is good, but you might want to say whether this leveraged an existing hyena reference database (and if so, mention the database coverage in terms of the current population). Perhaps mention in the supplemental methods.

- Lines 126 (and 241 etc...): In general, figures presenting actual data should be called out in the results.

- Line 138: I'd expand on the sample collection/field conditions. How were the samples collected immediately (without risk to researchers)? How was anthropogenic manipulation (Line 95) minimized in this study? Were cool packs in a cooler? Approximately how long did samples sit before reaching the field station?

- Line 148: Suggest "Microbial DNA extraction" to differentiate from the host genotyping?

- Line 149: Perhaps specify what the NucleoSpin Soil kit targets. I was unfamiliar and the Takara blurb only mentions bacteria, yeast, and fungi. Maybe include a reference for its use in helminths (e.g., doi: 10.1186/s13028-022-00624-3)

- Line 215: I interpret "repeatability" to mean the comparison of one sample repeatedly sequenced under equivalent conditions (technical replicates, e.g., PCR duplicates). Are you referring to technical replicates? It seems more likely that you are comparing longitudinal samples from hyenas that were sampled over time. This should be clarified given that these are very different research questions.

- Lines 221-222: I interpret "dyadic" to mean two linked individuals, and in this case, one could take it to mean mother-pup dyads. Do you mean "pairwise" here? State the specific beta diversity measure used (instead of "(distances)"). I'd reword or remove "(excluding within-sample comparisons)" since for a moment I thought you were referring to alpha diversity (but I think you mean comparing a sample to itself which would have BC=0)?

- Line 228: Will all readers be familiar with Stan? Perhaps say "in the Stan probabilistic programming language" and list the

R interface used (RStan, cmdstan?) + version

- Line 232: Looks like "the Sampler" should be lower-case when used as such https://cran.r-project.org/web/packages/brms/vignettes/brms_overview.pdf

- Line 244: Did you test for all interactions in the model before arriving at the final version? The reference you provided did find interactions between age and immune measures, but since here you are testing for influence of immune measures/environmental/social covariates on microbial diversity, you may reveal different interactions. Glancing at the code it appears that the immune/age interactions were manually entered but I don't see that the other variables were tested for interactions (i.e., separated via "+" rather than "**")

- Lines 248-250: Recommend shortening this section by adding `splitrule=variance` and `min.node.size=10` to the previous sentence (since they were the same for both immune measures) and then just specify the `mtry` for each

- Line 253: Describing taxa importance as permutation importance is a bit redundant and doesn't tell me much. Perhaps specify that this is a command argument (`importance="permutation"`) and explain what permutation importance means (something like "a variable's influence on the model's predictive performance")

- Line 253: Give the mean and range of samples collected for those 37 hyenas

- Lines 298-300: Contamination/final dataset. I realized contamination should be minimized due to the microfluidics approach but what was the percent of reads/ASVs removed by Decontam? Comment briefly on the ASV/sample drop out post-QC. On that note, 301 bacteria + 691 eukaryotes= 992 cASVs (not 999). Among eukaryotes 420 were fungi and 25 were known pathogens, what about the other 246 cASVs?

Reviewer #2

(Remarks to the Author)

In their introduction, the authors provide sufficient background regarding the state of research on the microbiome and lay out a compelling rationale for the study of wild animal populations as it relates to questions of host immunity and microbiome interactions. They proceed to analyze a longitudinal dataset of hyena samples to understand how host immune markers (f-IgA and f-mucin) relate to different microbial profiles, suggesting that microbiome composition is more similar in pairs of individuals with similar immune markers or age. Some microbes were further identified to be predictive of host immune markers, suggestive of stronger host-microbe interactions. Overall, the authors have conducted a novel study that contributes to the field; importantly, it looks at the microbiome more holistically by taking into account bacterial and non-bacterial members of the microbiome, and further does so in a wild population where multiple sources of natural variation influence the already complex interactions between host immunity and the microbiome. There are areas for improvement, specifically in considering the interactions between microbial groups and how those interactions may jointly influence immune markers or mask one another's effect on those markers. Addressing this area will strongly contribute to the importance of this work and further supplement the authors' argument regarding microbial cross-talk and the importance of a holistic approach to microbiome research. Some comments for the authors' consideration follow, with the aim of strengthening the overall presentation of the data and the analyses in this manuscript.

In lines 190-199, the authors describe their screening methodology for preparing sequence data for downstream analysis. While some reads discarded in this manner may not align or have biological relevance, it may provide a stronger argument to attempt an alignment of these ASVs to database(s) (i.e. SILVA 138.1) to check for potential matches and then provide further rationale for discarding these sequences. For example, some sequences may represent taxa that seem biologically unlikely to be incorporated into the hyena microbiome and should thus be discarded. As it currently stands, there is a likelihood that some rare taxa are not incorporated into the downstream dataset, which could disproportionately affect certain beta-diversity measures used in the analyses (Jaccard distances, as one example). The current approach biases the initial dataset towards homogenization rather than reflecting the desired natural variation that the authors initially note as a desirable feature of studying wild populations.

In lines 232-236, the authors describe testing the effect of individual repeatability on beta-diversity measures in a subset of samples (78/210) from a subset of individuals (37/165). It would be helpful to understand how these subsets were selected. Further, each individual is presumably duplicated in this subset of samples, but are some individuals replicated in triplicate given the total number of samples in the subset? Overall, it may be helpful to include a supplementary metadata table that clarifies the relationship between the 210 samples and 165 individuals, including frequency of repeat sampling for individuals particularly as it relates to this testing of repeatability on beta-diversity measures.

In line 239, the authors reference dyadic comparisons across samples, excluding within-sample comparisons. The number of dyads (19701) seems appropriate for 210 samples, but this cannot be validated without further detail on the nature of the sample collection, number of repeat samples per individual, etc. as described in the comment above.

Line 261 references Table S1, which includes two additional beta diversity measures that were used to analyze the effect of different factors as it related to microbial composition across different taxonomic groups. Table 1 in the paper includes a similar analysis focusing on Bray-Curtis distances. For ease of comparison, it would be helpful to include the BC-distance analysis in Table S1 as well alongside the other two metrics. Further, there is no discussion of these results from the other

distance metrics and how they support or diverge from the BC-distances; they seem to largely be consistent. By addressing all three and their similarity and dissimilarity, the authors can provide a more nuanced argument for the observed associations.

In lines 279-281, the authors reference using BLAST to "confirm" the taxonomy of ASVs of interest. However, in the provided table (S3), only 8/33 taxa are matched at 100% alignment, most to "uncultured bacterium" rather than a genus or other taxonomic identifier, which was the purpose of the BLAST. Softer language or a caveat regarding this confirmation may be more appropriate.

In lines 282-288, the authors describe a co-abundance network analysis identify associations between potentially ecologically relevant taxa. Does using all of the data (without the filtering down to 20% prevalence or higher) affect the results? Given the prior filtering step to remove artifacts and rare sequences, further filtering seems unnecessary. Sparsity of sequences does not inherently obviate ecological relevance, so relevant data may be excluded by this filtering - especially as this seems to have dropped the number of cASVs from 999 to 143. It would be helpful to either see a comparison of the analysis, with and without filtering, or for additional reasons to be provided for filtering down to 20% prevalence and higher.

In line 292, the authors reference sampling the intestinal community of 158 individuals; this seems to contradict the 165 referenced in lines 125-126. This 158 number is also referenced in the abstract in line 36, in the Figure 1 caption in line 310, and in the Figure 2 caption in line 337.

In discussing the association between host immune measures and microbial groups, the authors should consider controlling for similarity or dissimilarity of other microbial groups, which may more accurately attempt to capture the "cross-talk" between communities. As an example, the authors note alignment between bacterial and parasitic profiles (lines 419-420). Therefore, the analyses where the effect of a particular community (i.e. bacteria) on f-IgA or f-mucin are likely not entirely isolating that community; there is an entanglement between the bacterial and parasitic profiles, in this example, that may be falsely attributed to one or the other or resulting from some interaction between them. The authors might consider the following:

1. Noting the potential effects of this interaction in the results, which would require less effort.
2. Attempting to measure this interaction and including it in the model, which would require more effort but strengthen the manuscript and arguments therein. The same could be repeated for each pair-wise community interaction, as relevant.
3. Control for the interaction such that the resulting data only reflect, to the extent possible, the influence of just one microbial community, which would still require some effort.

This expanded analysis (options 2 or 3) would further strengthen the discussion on the co-abundance of these microbial groups.

While age, time, and parentage are all considered in the evaluation of the similarity of both host immune markers and microbial profiles, sex seems to be a known data element (except for those individuals who were too young) that is not addressed for its potential impact or lack thereof. This would likely only require a sentence to address and minimal effort if the data is already part of the existing metadata for these samples.

Further, is there any existing data from the long-term study of these individuals that would allow for a discussion of the impact of environmental effects, seasonal effects, or dietary effects? Including these data may be significantly more effort, but would complement existing discussions and observations of these trends in the literature and add to the comprehensiveness of the analysis in this manuscript.

Finally, is there existing data on the social interactions between individuals? Multiple publications already point to the effect of social interactions on microbial similarity, as the authors themselves cite, but there are novel data here regarding other microbial groups. A social network analysis, even with a subset of data, might further complement the existing findings if feasible. This would require significant effort, assuming the data exists but needs to be manipulated into the correct form for analysis.

Version 1:

Reviewer comments:

Reviewer #2

(Remarks to the Author)

The authors have made significant efforts to revise the manuscript as per reviewer feedback. Overall, the authors present a much stronger argument on the relationships between mucosal immunity and the gut microbiome, including both bacteria and eukaryotes. Changes to figures, explanation of sampling and the more detailed methods, and the added nuance throughout the text are notable edits. The authors are careful not to overstate their results, particularly given challenges in interpreting fecal-associated markers and microbiome data with well-characterized host-immune interactions in the small intestine. The authors' approach to labeling edits is also appreciated. There are some additional considerations, provided below, the authors might consider to further evaluate the manuscript prior to publication; these are largely minor as the authors have submitted an excellent manuscript for consideration.

Given the importance of age on beta-diversity (F2) and f-IgA and f-mucin (F4), is it possible to tease out how much of that

effect derives from comparisons between adult-juvenile pairs vs. other pairs? It would be important to highlight if there was a non-adult-juvenile age-related pairwise comparison that was driving these trends. Given the longitudinal data, are there any samples from juveniles who grew into adults across samplings that could be used as another subset for further exploring these trends? Similar to data in Figure S3, if there is information on age of sampled individuals, that may also be an informative histogram.

The new supplemental Figure S1 is generally clear in describing the methodological pipeline, removal and filtering of samples, and combination approach. Further, the primer table is a clear and useful resource to clarify the approach for the study design. The authors' broad targeting methodology for amplification and sequencing can result in multiple sequences from a single taxon, which is well-acknowledged. In Figure S1, the authors state "After combining ASVs likely produced from the same taxon,...", similar to lines 246-249 in the main manuscript. It is ambiguous if this combination is the step resulting in the cASVs (when multiple ASVs are included) or a separate step. If this is not a separate step, there may be concerns about how some measures, like Bray-Curtis distances, are calculated, as these measures rely on abundance counts, which could result in a bias towards taxa sequenced multiple times under different primers. In S1B and/or Table 1, it may be helpful to clarify the process for avoiding duplicate counts of taxa sequenced multiple times given the sequencing approach when creating cASVs and when calculating Bray-Curtis distance metrics. Given the general similarity of observed results with other beta-diversity metrics, this is likely a minor concern.

The purpose of Figure S2 seems to be a general indication of removed phyla due to filtering, but it is only limited to those taxa that appeared as singletons. Is there a reason to not include those taxa that were filtered for having less than 100 reads? This is addressed in the figure caption, but apparently not included in the figure itself.

Text edits:

Simple to single, plural ASVs (Line 257) - "For simplification purposes, we use the term cASV regardless whether it refers to a single or combined ASVs."

E to D (Line 504) - "... cASVs for predicting f-IgA and f-mucin. D) Top: Co-occurrence network of 1597 taxa..."

Dear reviewers
Communications Biology,

We thank you for your helpful and constructive comments and for the opportunity to submit a revised version of our work. We have carefully considered all comments and addressed them in a substantially revised version of our manuscript. In this document, we provide detailed answers to the questions and concerns raised. In addition we have applied changes related to style and formatting as per guidelines. Please see below, after a ">>", for a point-by-point response to all comments. All line numbers refer to the revised manuscript file.

We also include a revised version of our manuscript, in which the changes made are highlighted and linked to the comment number and reviewer. We hope this helps to assess our implementation of the required changes

Note that the line numbers mentioned by the first reviewer do not always match the lines in the pdf proof. We have thus rephrased the words or sentences the reviewer was likely referring to in our responses, to avoid potential confusion.

We hope that our extensive revisions addressed the concerns raised by you and that you will find the revised manuscript suitable for publication in *Communications Biology*.

Sincerely,
The authors

To sum up:

>> In order to address the questions raised we have conducted a profound revision of our manuscript, as mentioned above. We have provided additional details in numerous places as requested and clarified our code to ensure replicability. In addition, we conducted a new analysis, addressing the concerns for batch effects and other co-founders. We briefly highlight here the major changes done in response to the requests:

- 1) We have changed the quality filtering that was previously removing all ASVs with low prevalence and relative abundance (<1% prevalence and 0.005% relative abundance) to removing only ASVs present in one sample (singletons).
- 2) A new figure S1 details all the preprocessing steps and highlights the initial number of samples, individuals and the step in which samples were removed, and also provides details on the normalisation used.
- 3) The models analysing the β -diversity measures now include, in addition to the previous predictors, the sequencing batch and sampling seasonality. In addition, we have added a new model for all juveniles in which we include the effect of sex as a predictor. We do this only for juveniles because all adults in this study are females.
- 4) When analysing specific components of the microbiome (bacteria, fungi and parasites), we included as a predictor the similarities of the other components, to account for possible interactions among microbiome components.
- 5) The random regression models include, besides the relative abundances of all taxa, the age at sampling, season, clan and genetic mother, to disentangle the effects of microbiome and host-related effects and host-environment.

- 6) The network estimation included all taxa, instead of the taxa with at least 10% prevalence.

Reviewers' comments:

Reviewer #1 (Remarks to the Author):

Manuscript summary: In this study, the authors investigated immune measures (IgA and mucin) and used a multi-amplicon approach to profile bacterial, fungal and eukaryotic parasitic communities in 199 fecal samples collected from 158 wild hyenas over a ~14 year period. This work follows up previous studies that demonstrated individualized microbiomes in this population (influenced by factors such as age and prey availability), and a positive correlation between fecal IgA and mucin concentrations and *Ancylostoma* egg burden (also correlated with age and host survival). The goal of the current manuscript was to determine the relationship between immune measures and gut microbial community composition. The hypothesis was that host-associated microbial community composition was strongly associated with fecal IgA and mucin levels. The authors used probabilistic models and machine learning in their analysis. Their main findings were: 1) immune measures predicted microbiome similarity in an age-dependent manner; 2) immune measures were more strongly correlated with the bacteriome than parasites and even more so than fungi and 3) microbiome composition could be used to predict immune levels. Specific taxa driving this association were identified, including *Ancylostoma*.

Overall impression of the work: This study leveraged a dataset that was impressive both in terms of sample size and study duration. Given the recent research coming from the same group of researchers that has shown this hyena population has individualized fecal microbiotas and fecal IgA/mucins levels/parasite burdens associated with survivorship, it seemed fitting that this study bridge these factors by linking IgA/mucin to microbial diversity. That said, I found it hard to keep track of which taxa were being targeted, especially because certain results highlighted taxa that I had assumed were off-target (vertebrate DNA and organisms unclassified at the level of domain/phyla). Admittedly, I have not worked with multi-amplicon microfluidic PCR myself, but without a clear description of data normalization and consistent data subsetting, I was not convinced that the relative abundance of certain microbial groups was directly comparable. I also felt that the bulk of the figures revolved around the Bayesian probabilistic model and pairwise comparisons of both diversity measures and covariate measures, which are not intuitive for the reader. I believe that the incorporation of more straightforward visualizations along with such analyses will be more effective. Overall, I think this manuscript has great potential but will require some work to clarify the methods and better visualize the results.

>> Thank you very much for your very careful and detailed evaluation of our study. We appreciated that you found our work of great potential.

As suggested, we clarified how we have normalised and subsetted our data and provided an explanation on the interpretation of amplicon sequencing, in order to address your comment related to the comparison of the abundance of microbial groups. We also revised our methods in numerous places to clarify our approach. Second, we redesigned our figures to

visualise and interpret our results in a much simpler way and have added supplementary figures with alternative visualisations and analysis.

We appreciated the time and effort you invested, and the important points that you raised. Addressing them allowed us to substantially improve our manuscript.

COMMENTS TO THE AUTHORS:

Major comments:

Data reproducibility & confusion about methods: The following issues made it difficult for me to fully understand and evaluate what was done in this analysis.

R1C1 Code: Thank you for providing code, but it looks like `rFit_IgA.rds` is missing from the `tmp` folder so I was unable to reproduce the latter part of the analysis. I'm also unable to link the sequences to the ASV IDs used in the manuscript when starting with the `fPMS.rds` phyloseq object. (Minor note- for the next submission, please organize the code according to the figure numbers so chunks are easier to locate. For example, the code refers to "Fig5" but there is no Fig 5.)

>> Thank you for going through our code and for identifying these issues. We have added `rFit_IgA.rds` to the `tmp` folder. Please note that some files stored in the `/tmp` subdirectory might be missing due to size constraints. This is the case for the `tmp/se.fnet.rds`. In such cases, the chunk of code needs to be run, e.g. by uncommenting the line that is creating the object.

The sequences can be linked to the ASV IDs at the random forest and network sections. The reason why we change the name in the first place is for visualisation purposes in the figure 4, however, we prefer keeping the sequence as the sequence identifier throughout the analysis. The ASV IDs are simply combining a sequential number (e.g., ASV1, ASV2, ASV3) with the corresponding genus name from the taxonomy table. For the taxa that are highlighted in figure 4, we further performed a blast search of the sequence based on the hit quality, and reported in additional file 2.

We also substantially re-organised our code and verified that all remaining comments refer to figures that are present in the manuscript.

R1C2 Primer selection: I'm confused about which taxa were targeted in this study. The primer table is missing some information and some primers aren't mentioned in the manuscript (i.e., CytB, HSP70, tRNA and COI). Were all these primers used in this study? Was qPCR done? Was host DNA targeted in addition to microbial? The manuscript divides the data into 3 groups: bacteria, fungi & parasites, so I would tidy the primer table and organize it as such. In addition to the broad "Target" column, add detail what was targeted by specific primers at a finer level (e.g., protozoa/helminths/fungi within Eukaryota). Add citations to back up primer choices. This should be provided as a supplemental table/file and I would consider adding a simplified table to the main manuscript. Below are some related comments regarding the associated text that may clear up confusion:

>> We have updated the primer table (additional file 1) to add the missing information and citations of all primers used in this study. We now mention all the genes targeted in the main text (lines 202-206), and refer to supplements file - list 1 for further detailed information on each primer pair. Although each primer pair will have biases towards specific taxonomic groups, we employed a combination of primers that allow us to amplify a broad range of taxa and we also include a few more target specific primers (eg. for apicomplast genes that target *Eimeria*). Given the broad-amplification range of each primer, we did not employ any other method for target-specific identification, like qPCRs. Apart from primer pairs that target bacteria, primer pairs targeting eukaryotes do not exclusively target fungi or parasites (even within apicomplexa, many taxa are not considered parasites, see discussion on comment R1C47). Because of that, the “Target” column is broad, as the reviewer pointed out. Additionally, primer pairs that target eukaryotic DNA also target host and diet DNA in addition to microbial (and macroparasites) DNA. We recognize it as a drawback related to nonspecific amplification. However, the aim of our study was to understand the role of multiple species and thus the use of broad-range primers provides enough robustness to identify more Genera (figure R1 below) and cover most of the community diversity. The latter corresponds to an ongoing work aiming to compare multiple primers.

Fig. R1: Multiple marker genes and multiple primers increase the detection of parasites (A) and fungi (B) in gastrointestinal communities. Number of different taxa identified at different taxonomic levels by different primer pairs. Cumulative lines represent each taxonomic level, dots represent a primer pair combination and color the marker gene targeted by the given primer pair (*Unpublished*).

We acknowledge that target taxons are not always what is “hit”, and we find this topic important and interesting. We, however, feel that this topic is beyond the aims of the current study and should be developed in a dedicated study. Indeed, we are at the early stages of preparing a manuscript that compares primer specificity, with a specific focus on targeting parasites, by using a similar multiamplicon technique to the faeces of different mammalian species within natural populations.

R1C3 Lines 187-191 are redundant. Remove the first sentence and clarify the targets of each (e.g., “the UNITE database for fungal ITS...”).

>> The reviewer refers to the lines starting with: “Sequences targeting the 18S, 16S, 28S and ITS rRNA ...” We have removed the first sentence and added the target of each gene (see lines 237-242 and the following quote) “We used the SILVA 138.1 SSU Ref NR 99 to classify eukaryotic 18S and bacterial 16S rRNA gene sequences, the SILVA 138.1 LSU Ref NR 99 databases for eukaryotic 28S rRNA gene sequences, and the UNITE database for eukaryotic ITS rRNA gene sequences. All other sequences from targeted regions without publicly available curated databases were classified against sequences downloaded from NCBI using RESCRIPt”.

R1C4 Line 205 refers to “four groups of cASVs” but only three are listed. Plus, are they all really “cASVs”? For example, only one primer pair was used to target 28S, so wouldn’t these sequences just be “ASVs”? If both cASVs and ASVs were used this should also be specified in the primer table.

>> Thank you for spotting this. We have changed “four” to “three” groups (lines 260-263 - “We investigate the intestinal community by considering all taxa detected within the samples, regardless of the taxonomic annotation (overall) and by decomposing the community into three groups of cASVs:). For clarification, we use primer pairs that target a very broad taxonomic range (also see our responses to other comments like R1C2). For instance, the primer pair targeting 28S rRNA gene targets Eukaryotes, but so do the primer pairs targeting 18S rRNA gene. This means that primer pairs targeting 28S and 18S rRNA genes can potentially amplify the same taxon, as seen for e.g. *Eimeria* spp. in Ferreira et al. 2023, figure R2 (below). In this figure each circle represents an ASV resulting from primer pairs targeting 18S rRNA gene (yellow, green and blue) and 28S rRNA gene (violet).

In this example, and based on this co-occurrence network and phylogenetic analysis (figure 3 and 4 in Ferreira et al. 2024, not shown here), the result is one cASV for *Eimeria ferrisi*, one cASV for *Eimeria falciformis* and one ASV for *Eimeria vermiformis*. Both cASV from *Eimeria ferrisi* and *Eimeria falciformis* contain ASVs amplified by primers targeting 28S and 18SrRNA genes. As the reviewer points out, and shown in this example, not all ASVs are indeed “cASVs”. We have clarified this in the lines 246-249 and 254-257. We have included figure S1B that further explains the cASV approach.

Figure R2: Fig. 5 from Ferreira et al. 2023. Co-occurrence network of ASVs annotated as three *Eimeria* species based on phylogenetic analysis. Nodes represent all ASVs annotated to *Eimeria* sequenced with a multiple amplicon approach. Node size reflects the frequency and relative abundance of each ASV: the relative abundance after total sum scaling within each amplicon, and summed across amplicons for each sample. Edges represent significant Pearson correlations after adjusting for multiple testing. Green edges mark positive correlations and red edges mark negative correlations.

R1C5 Lines 205-211: Just list the three broad categories (bacteria, fungi & parasites). Why are specific phyla are listed here? If these were the targets, include that information in the primer table. If these are lists of the successfully sequenced phyla, this information should be in the results.

>> These are the lists of successfully sequenced taxa for each group. We have moved these lists to the results, see lines 366-373 (“301 cASVs originated from the bacteria domain, including the phyla Firmicutes, Bacteroidota, Campylobacterota, Cyanobacteria, Proteobacteria, Planctomycetota, Fusobacteriota,....”) and refer to the three broad categories lines 260-263 (“We investigate the intestinal community by considering all taxa detected within the samples, regardless of the taxonomic annotation (overall) and by decomposing the community into three groups of cASVs: 1) bacteria domain, 2) fungi kingdom; and 3) eukaryotic parasites”).

R1C6 Non-microbial reads and unclassified taxa: I’m not sure why the authors decided to retain vertebrate DNA in the analysis since the focus was on the microbiome. They also retained unclassified ASVs. Since these are typically filtered out early in the analysis, I suspect removing these ASVs may significantly affect the results.

>> After the quality filtering aimed at removing lowly prevalent ASVs, we retained all ASVs regardless of their classification. We did so to not potentially lose ASVs that are important within the gut community but not yet present in the databases, as was done in several other

studies^{1,2}. Indeed, the models considering the overall microbiome β -diversity were based on all taxa, regardless of taxonomic classification. This approach includes unknown symbionts (regardless of their functional role in the host) but also transient taxa and elements of the intestinal ecosystem, such as host diet and even host DNA.

In addition, we restrained our next analyses to known specific groups of the microbiome, in which vertebrate DNA and unclassified DNA was not included, by modelling only bacteria, fungi and eukaryotic parasites. The effects of IgA and mucin, our main results, are robust to whether we model all taxa detected in our samples or defined taxonomic groups (bacteria, fungi or parasites), see table S2, with the exception of the non-significance of IgA on fungi β -diversity. We clarify both approaches in the text, line 260-263 and quoted there “We investigate the intestinal community by considering all taxa detected within the samples, regardless of the taxonomic annotation (overall) and by decomposed the intestinal into three groups of cASVs: 1) bacteria domain, 2) eukaryotic parasites, and 3) fungi kingdom.”

R1C7 Off-target amplification: Was the hyena host DNA specifically targeted here? Or was this host contamination as is common with certain primers? Most microbiome studies aim to avoid amplifying host DNA so that a larger proportion of the reads are microbial targets. Some even use PCR blockers to avoid host contamination. If the host DNA was of interest, wouldn't quantification be more important (i.e., qPCR, ddPCR, WGS)? If the goal of using host DNA was to examine inflammation, wouldn't immune assays or even stool blood content be more appropriate?

>> Thank you for bringing this up. Host DNA was indeed targeted because we used “universal” or “generic” primers that target eukaryotes in the broad sense, see reply to comment **R1C2**. Host contamination (off target amplification of host DNA) is a common issue when sequencing low microbial load samples, like skin, or when targeting 16S rRNA gene of faecal samples, with the aim to characterise the bacteria component. For the latter case, since taxonomic annotation in such studies is achieved using curated databases of bacteria sequences, e.g. SILVA SSU, host DNA is usually unclassified and all unclassified reads are removed. In our study we aimed at characterising taxa beyond bacteria. Although we did not have primers designed to amplify specifically host DNA, we do not consider host DNA as a contamination in a strict sense, since primers targeting eukaryotes will also target vertebrates.

This study did not aim to measure host DNA specifically. Indeed, host DNA was highlighted in the exploratory part of this study (co-occurrence network and random forest regression results of previous version), as we did not have an *a priori* hypothesis about the relationship between host DNA and faecal mucin and IgA.

We now explicitly mention in the introduction that, besides the hypothesis driven analysis, we also include an exploratory analysis of co-abundance networks and predictive taxa based on the random forest regression, lines 126-127, quote “We further explore the inter-species interactions.....and highlight the taxa with the strongest links to immune measures.” We also did it in the results line 519 “We further explored inter-taxa interactions with a co-abundance network of all cASVs. ”We also repeat in the discussion “We explored which taxa showed the strongest associations with immune measures.”line 549.

¹ Ley et al., ‘Evolution of Mammals and Their Gut Microbes’.

² Zhang and Chen, ‘Phylogenetic Analysis of 16S rRNA Gene Sequences Reveals Distal Gut Bacterial Diversity in Wild Wolves (Canis Lupus)’.

R1C8 Furthermore, there I don't see a "Crocuta" designation in the tax table and the only cASV from Vertebrata. This BLASTs to many mammalian species (e.g., 1st hit is an Etruscan shrew). How is it known that this is hyena DNA and not prey DNA? What is more confusing is that at line 486 in the discussion the authors state "our results indicate that prey DNA is negatively correlated with host DNA". I cannot find any analysis to support this claim. How could this comparison be made with only one cASV corresponding to Vertebrata? Several cASVs correspond to arthropods, but wild hyenas must be eating vertebrates as well.

>> Thank you for identifying this lack of clarity, and apologies for the confusion. The cASV classified in previous figure 4 as *Crocuta crocuta* was unclassified at the domain level. Only when blasting each sequence from the top 20 most important taxa in the random forest regression model, did we conclude that this particular ASV was likely from spotted hyenas. The list 2 in the supplements has the sequence, the ASV ID to correspond to the figure and alignment information of the first hit on the NCBI BLAST. We apologise for the confusion that we consider to emerge from changing the cASV sequence to an ID for easier visualization only in the last stage of the code, which regrettably, the reviewer could not reproduce. We have added more explanations about the code and the repository structure. We also more clearly communicate which cASVs taxonomic classifications were changed as a result of our BLAST search. Lines 328-329 and the following quote "We have adjusted the taxonomic annotation accordingly as shown in supplementary data file 2."

The results backing our claim that "our results indicate that prey DNA is negatively correlated with host DNA", referred to the co-abundance network in which host DNA (previous ASV 779 *Crocuta crocuta*) was negatively associated with previous ASV 774 *Connochaetes taurinus*, previous figure 4E, node colours green and purple. We have added host and environmental predictors to the random forest models and interestingly, host and prey DNA no longer belong to the top 20 most predictive taxa of IgA and mucin (fig.4).

As for the ASV classified as vertebrate that the reviewer refers to, we agree, we do not know which vertebrate it comes from. As this particular ASV was not within the top 20 most predictive taxa of immune levels, we did not attempt to further investigate its taxonomy or interpret its position within the co-abundance network.

R1C9 Unclassified taxa: What was the rationalization behind keep taxa with unclassified kingdoms and domains? Typically, taxa that aren't assigned to the level of phyla are removed prior to the data analysis since they may be spurious. This is especially curious considering many of these taxa were identified in the machine learning model. I appreciate that the BLAST table was provided, but the fact that so many hits were to unclassified bacteria or even "unclassified organism" makes me think they should have been filtered.

>> There are several reasons to justify our choice not to remove unclassified taxa, i.e. "dark matter" from all the analyses in this study. The first reason is technical, as we are targeting marker gene loci that are not commonly used and thus under-represented in available databases. The second reason is biological, and partially explained in comment **R1C6**. Since we are investigating the intestinal microbiome of a non-model and non-human organism, the available knowledge of the taxa present in such communities is currently very limited and thus the "dark matter" comparatively larger. For instance, we did a blast search for ASV_8_Unknown_genus_in_Unknown_domain, (supplements list 2), and the top hits were uncultured bacteria isolated from the faeces of wild wolves (accession FJ978585.1), and

captive cheetahs (EU459381.1 and EU773734.1). Similarly, the ASV_477_Unknown_genus_in_Bacteria (now ASV_685) top hits on NCBI BLAST search were uncultured bacteria isolated from the faeces of snow leopards (KC245386.1) and cheetah (EU773856.1, KF909824.1). We take this as an indicator that these bacteria are symbionts in wild mammals, but not yet classified. Thus, in the case of understudied wildlife populations, we argue that keeping taxa with unclassified kingdoms and domains is a more robust approach as it avoids the risk of removing important but still unknown symbionts. We have added the isolation source on additional file 2, of the top hit for the most predictive taxa of IgA and mucin.

R1C10 Merging ASVs: Line 201 refers to the Ferreira et al., 2023 manuscript for the detailed methods of merging ASVs. However, it appears that in that study the tree was trimmed down to just *Eimeria*. The current analysis seems more complicated since multiple primers were used on multiple targets. Perhaps another reference is missing from the claim in line 202 “is now extended to all annotated genera” that contains such a workflow? If not, I recommend rephrasing that to something like “and is extended to multiple microbial targets in the current study” and expanding on the workflow description (e.g., including supplemental trees, perhaps before and after merging into cASVs).

>> The approach used in the current study is indeed based on the Ferreira et al. 2023 study that tested different normalisation techniques and the cASV approach for detection and differential quantification of 3 different species of *Eimeria*, against the golden standard measures (qPCR), in both controlled laboratory infection and natural infections. It has been extended to all taxa (bacteria and eukaryotes) in Ferreira et al 2024 DOI: 10.1098/rspb.2024.1970, which applies the method to multiple microbial targets (bacteria and eukaryotes) and includes further analysis on the congruence between the cASV approach and the phylogenetic analysis of Oxyuridae. We have reformulated this sentence (line 254-255, here quoted “This has been previously accessed for *Eimeria* spp.^{83,84}, and Oxyuridae, and extended to multiple microbial targets⁸⁴”. We include a visualisation of the approach in figure S1B.

R1C11 Normalization: How was this data normalized? The all-in-one approach of using the Fluidigm Assay has obvious benefits, but one would assume that different primers have different ideal conditions, and without control over individual reactions, bias might be introduced. In Ferreira et al., 2023, qPCR was used. Was it used here as well? Even so, the previous study was only concerned with *Eimeria*, so how did the authors in the current study account for multiple kingdoms of life? For example, in Fig 2A,B the data from eukaryotes is lumped together, but given it spans 18S, 28S and ITS, I’m not sure that this is appropriate (see comments on Figure 2).

>> As the reviewer points out, we use the same methodology used in Ferreira et al. 2023 that has extensively tested different normalisation techniques against the golden standard measures (qPCR), in both controlled laboratory infection and natural infections of *Eimeria* and has been also applied in Ferreira 2024. Here, and similar to the previous studies, we normalised by total sum scaling for relative abundances of each amplicon (product of one primer pair) before collating all products from each amplicon into one “phyloseq” object. We have clarified this in the text, lines 228-231 and here quoted “Each amplicon in the multi-amplicon datasets was individually filtered and normalised by total sum scaling for relative

abundances and then all products were collated into an “phyloseq...”. We agree that different primers have different biases towards different taxons in estimating taxa abundance. To this end, no “perfect” primer has been identified that targets all taxons within multiple kingdoms of life. The multi-amplicon approach gives us the opportunity to capture a larger diversity of eukaryotes and prokaryotes than a single-amplicon approach, especially for complex environments like wildlife microbiomes (see comment R1C2 and figure R1). Given that each primer pair has its own “blind spots”, we argue that targeting 35 different gene loci, with carefully selected primer pairs, minimises biases towards a specific taxonomic group. In addition, the selection of these primers was done also based on their suitability to efficiently amplify using the Access Array system and the same amplification program for all primer combinations.

The resulting taxa abundances of both single and multi-amplicon are compositional, meaning that their abundances estimates impact each other. Regardless of the inherent biases, e.g. associated with unequal amplification, insufficient coverage of primers, these methods are nevertheless useful in determining the changes in the relative abundance of taxa within samples. Thus, the interpretation of results must always take into account that the abundance estimates of each taxa within a sample are relative and not absolute. Given this, we refrain from comparing taxa abundances within a sample (e.g. taxon A has higher abundance than taxon B), and instead focus on the differences between the samples (sample A is similar/dissimilar to sample B - β -diversity). We include a short explanation in the supplementary materials.

R1C12 Data subsetting: From the code it seems that parasites and fungi were subset down to select taxa, while bacteria were not. What was the rationale? Isn't the goal to get a wholistic view of the microbial community? If the data was not normalized and then certain clades were subset even further, I'm not sure it is appropriate to compare across groups. For example, one main claim is that bacteria had a stronger association on immune measures than parasites and fungi, but perhaps that is because the entire bacteriome was considered versus just a subset. Line 301 says data was subset to “known parasites”- what about other microeukaryotes that might correlated with ill health? Also, how were there pathogens selected? Toxoplasma, a known hyena parasite, was left off this list but was detected in the study. What about microeukaryotes that may have beneficial roles (see: <https://www.frontiersin.org/journals/microbiology/articles/10.3389/fmicb.2023.1123513/full>)?

>> We apologise for a potential misunderstanding here. We included all taxa annotated as bacteria domain, all taxa annotated as a fungi kingdom and similarly, all known eukaryotic parasite taxa. The reason why we selected the fungi phyla individually in the code was simply because we did not have the fungi kingdom information in the taxonomy table. Therefore we manually inspected all phyla and selected the ones belonging to the fungi kingdom. To be clear, all taxa belonging to the kingdom fungi were included.

In contrast to bacteria or fungi, the parasite component is not a taxonomic group. Instead, it is classified according to its functional role for the host. When we referred to “known parasites”, we meant the only unambiguously functional annotation we can perform in our study system, based on previous work. We had performed faecal egg counts in previous studies (Heitlinger et al. 2017, Ferreira et al. 2021, Ferreira et al. 2019), which indicated that the taxa identified are actual intestinal parasites completing their life-cycles in the intestine. Thus, we only considered parasite taxa that are plausible intestinal parasites in hyenas. We inspected all

taxa from the phyla nematoda (annotated as nematzoa in this study), platyhelminths and apicomplexa, and selected all likely parasite taxa.

Toxoplasma is a known parasite in hyenas, but it does not reside in hyena intestines (10.1016/s0020-7519(00)00124-7, 10.1016/j.ijppaw.2018.12.007, 10.1038/s41467-021-24092-x). We indeed have ASVs that are annotated as *Toxoplasma* but we consider that to be either *Toxoplasma* DNA acquired from eating the cyst within an infected animal, or a misannotation, since *Toxoplasma* only divides in the intestine of cats.

Unfortunately, our knowledge is restricted to eukaryotic parasites. We lack knowledge to classify other taxa that might be detrimental or beneficial to the host. Identifying taxa that are associated with health or even fitness benefits would be the first step in this direction.

In the manuscript, we specified which taxa were found for each group and in response to **R1C5**, have moved this part from the methods to the results (see lines 361-373 and here quoted “The overall intestinal community was composed of 1597 cASVs deduced from 4706 ASVs across 35 different amplicons. 1183 cASV originated from eukarya, 407 cASVs originated from **R1C5** bacteria,”

Taxa were normalised per amplicon (see response to **R1C11**).

We acknowledge that different microbiome components analysed here vary in the number of taxa, which likely influences the association with immune measures. We highlight this in lines 542-544 quoting “Known eukaryotic parasites in this study comprise fewer taxa than bacteria and fungi and were still strongly associated with both immune measures, highlighting the strong relationship between parasites and host immunity.” We soften our language when comparing effect sizes between the groups (e.g. lines 539-540).

R1C13 Sample metadata: It’s unclear how many runs were performed or whether data was analyzed for batch effect. The metadata should reflect which run each sample came from as well as the negative controls. DNA concentrations used in Decontam (are these from Qubit or qPCR?) should be included as well. Were technical replicates included (i.e., same sample, different run/reaction)? If so, these should be specified in the metadata (even if they weren’t all used in the main analysis)

>> Thank you for bringing this to our attention. We have redone all the models to account for possible batch/run effects (revised tables S1, S2, S3). We emphasize it on line 283 “We also accounted for possible sequencing batch effects (same vs different)”. We have added a new figure (figure S1) that details the bioinformatic pipeline, including how many runs and batches, and technical replicates were included. Batch and runs are specified in the metadata and explicitly annotated in the code. We also clarified that DNA concentrations came from NanoDrop. Lines 225-226 quote “...based on DNA concentration estimated spectrophotometrically with NanoDrop (Thermo Fisher Scientific, Germany).”

R1C14 Confounding age & sex covariates: Why were only adult females sampled? The rationale here should be described. Was it not possible to sample adult males? From Ferreira et al., 2021: “Fecal concentrations of IgA, IgG, and mucin increased with Ancylostoma egg load and were higher in juveniles than in adults. Females had higher mucin concentrations than males”. The fact that all adults in this study are female makes it

more difficult to parse out the influence of sex, age, and immune measures on the microbiome.

>> Thank you for reading our previous work. It is possible to sample adult males, but as female spotted hyenas are philopatric and males disperse when they reach adulthood (as written lines 104 -107 “Spotted hyenas (hereafter ‘hyenas’) live in stable social groups (clans) with members varying in their social status within a linear dominance hierarchy, in which adult females and their offspring socially dominate immigrant males ⁴⁹⁻⁵¹, many research questions focusing on processes occurring in adults are restricted to females in our population (see eg 10.1038/s42003-024-05926-y, 10.1111/1365-2656.13785). The great majority of adult males in our study clans are immigrant males that were born in non-study clans and for which the rich life history data that we have for females (birthdate, early life conditions, maternal effects, ecological conditions etc) is not available. We thus chose to focus our study on adult females, as we have more information about the experiences they have encountered throughout their lives and that may affect the measured variables. We have clarified why only adult females were sampled in the methods and results, lines 142-144 and 359-360 quotes “We limited our focus to cubs of both sexes and adult females as most adult males tend to disperse upon reaching adulthood, making it challenging to monitor them throughout their entire lifespan “ and “ All adults were female (61 samples from 58 individuals), juveniles were of both sexes (138 samples from 41 males and 81 females).”

We have included a new model restricted to juveniles that investigated the effect of sex on the β -diversity measures, and found no effect, Table S3 and 441-443 “within juveniles, we find no effect of sex on the overall microbiome composition and also on the bacteria, parasites, and fungi (Table S3).”

R1C15 FIGURES: In general, I felt that there was redundancy in the figures, while some important concepts were not visualized. Specifically, Table 1 conveys the same information as Figs 2E,F and Fig 3A-C. The goal of the paper is to show an association between immune measures and microbial diversity, yet we don't see plots of alpha and beta diversity measures over IgA/mucin levels aside from the pairwise comparisons (which can be hard to digest). In addition, although temporal distance between samples is significant for all microbial groups in Table 1, only age is further analyzed.

>> Thank you for your feedback about the figures and tables. We have now re-designed all figures and created new supplementary figures. We have moved table 1 to the supplements. Revised figure 2 shows all host and environmental variables effects on the β -diversity based on all taxa (overall model). This includes temporal distances. Figure 3 shows the specific effects of immune measures on β -diversity on the overall, fungi, parasites and bacteria communities, and how the interaction of age differences and immune measure distances affect β -diversity.

We added ordination plots with vectors indicating the correlation of immune measures, figure S4. We did not include any alpha-diversity analysis in our manuscript as we did not have clear predictions of the effects of immune measures and we consider β -diversity more informative.

FIGURE 1:

R1C15- I don't find the visualizations of the pairwise distances in B,D and F that interesting, they could go in the supplemental. Instead, I'd rather see the histograms for the clans, social ranks and year of sampling from Fig S1 since these are the covariates used in the model in Table 1.

>> We agree. We have moved all histograms of samples/individuals to figure S3, and removed the histograms of the pairwise distances as they are not informative.

R1C16- I liked the hyena icons at first, but now feel like they are distracting. This is especially true for 1B since according to the dyad.data.rds there aren't any adult pairs at Age distance=0 (only juvenile-juvenile pairs). I think it's best to just note "more similar in age" or "similar f-IgA levels" with an arrow pointing towards "dissimilar ages" or something along those lines...

>> We have removed the hyena icons from the figures. We no longer show the histogram of age differences.

R1C17 Legend: The abbreviation RU is defined, yet I do not see it used in the figure. I believe "immune measures levels" should be "immune measure levels".

>> Thank you for spotting these issues, we have corrected them in the new version, figure S3.

R1C18 1A: This should be stretched out or divided into hyenas sampled once and those with repeat sampling because right now it is hard to see what is happening up top.

>> Thank you, we have changed the figure so that each dot does not overlap, revised figure 1A.

R1C19 1B: I'd label the x axis "Pairwise age distances"

>> No longer applies.

R1C20 1D: Incorrectly labeled 1E in legend. I'd label the x axis "Pairwise f-IgA distances" or "dissimilarities"

>> No longer applies.

R1C21 1E: Incorrectly labeled 1D in legend

>> No longer applies

R1C22 TABLE 1: First off, I'd start with the relative abundance plots (Fig 2A-D) and then show how these communities are associated with immune measures. If using this table (instead of Figs 2E,F & Fig 3A-C), I'd clean it up a bit by making it more obvious which are significant (like the supplemental table) and adding labels on the side to delineate the immune measures, social/environmental covariates and interactions.

>> Thank you for these suggestions. Table 1 was moved to the supplements, see reply to **R1C15**. Revised figure 1 shows the sampling (including repeated sampling) and the microbiome composition.

Figure 2 shows all effects on β -diversity (overall microbiome similarity). The reader can compare the effect sizes of the various host and environmental predictors. We added the labels on the side to make it clear which are host, social and environmental factors, as suggested.

Figure 3 shows the specific effects of immune measures on the overall microbiome and the fungi, parasites and bacteria groups. Figure 3A and B also have the immune measure coefficients of the overall model, however this repetition aids to compare among microbiome components (bacteria, fungi and parasites).

R1C23 Shouldn't a sex:mucin interaction be added given previous findings (see above)? Perhaps you could repeat the analysis on just juveniles to investigate sex.

>> Thank you for this suggestion. We do not include sex in the main model because all adults are females. As suggested, we conducted a new analysis restricted to juveniles that includes the effect of sex, table S3. We report no sex effects on the overall microbiome in juveniles, lines 441-443 "within juveniles, we find no effect of sex on the overall microbiome composition and also on the bacteria, parasites, and fungi (Table S3)".

.

R1C24 TABLE 1 and Figs 2E,F & Fig 3A-C: How does one interpret these results in plain terms? Most tables like this list the covariates as predictors, not the pairwise distance between covariates. And is there directionality associated with +/- values? That should be clearly labeled. The figures are also confusing because the x-axis is labeled with the covariate distance, but I believe it should be "BRM Estimate size". For example: Fig 2E has the x-axis title "f-IgA distance" yet the values are all negative and the f-IgA distances shown in 1D are all positive.

>> Thank you for this comment. We interpret these results as, among compared samples, there is an increase in microbiome similarity with decreased immune measure distances. In other words, two samples with more similar IgA levels also have more similar microbiome composition. The +/- indicate the direction: increase/decrease in similarity. We have adjusted the labelling of figure 2 and 3 for easier interpretation.

All distances are positive, ranging from very similar (distance of 0) to very different (distance of 1). The effect of these distances on microbiome similarity might be positive (increases similarity composition) or negative. We have revised the entire result section to improve clarity and interpretation of these results. See, e.g. lines 394-399, figure 2 and 3.

R1C24 FIGURE 2: Fig 2A-D feel like one figure about relative abundance, while E,F pertain more to Table 1.

>> As we re-designed our figures this no longer appears.

R1C25 2A,B: Again, for the reasons discussed above I don't think it is appropriate to plot these microbial groups together. This is especially true in B where the relative abundance adds up to 100% (which is arbitrary). At least split these up into parasites and fungi.

>> in the revised version, we now include a detailed description of the cASV generation procedure in the supplements file. The aim of the revised Figure 1B-E (formerly Figure 2A, B) is to visualise the number and relative abundance of cASV for each prokaryotic or eukaryotic phylum. We acknowledge the suggestion to divide the relative abundance of eukaryotes into individual panels for parasites and fungi. However, we did not implement it as our aim was to contextualise the overall eukaryotic composition used in the statistical modeling. Bacterial and eukaryotic composition are independently assessed. We used primers that mainly target bacteria and other eukaryotes, so the relative abundances of eukaryotes are not influenced by the relative abundances of bacteria.

R1C26 2E: These values (Overall, Parasite) don't match the table. Your code does provide the correct version, however.

>> Thank you for spotting this! We have corrected the figure, now revised figure 3A.

R1C27 2E,F: If you keep these I'd make the x-axes consistent so the differences are more clear.

>> We agreed and did so.

FIGURE 3:

R1C28 3A-C: I prefer table 1 since it is more obvious which are significant. If keeping these plots, I'd lighten the bars that cross 0 to highlight the significant groups

>> Thank you. We have removed these specific panels. We have improved the label of revised figure 2 and 3A.

R1C29 3D,E: Can you clarify what was done here? Did you simply take the model from Table 1 and enter either AgeDistance=0 or 1 and then generate predictions along set intervals from f-IgA 0 to 1 (similar to dissimilar pairwise f-IgA comparisons?). This is hard to interpret without the number of samples and individuals used. Again, there are no adult-adult with AgeDistance=0 in the dyad.data, so 0 is only juveniles (both are shown in the legend). Also, there is only one pairwise comparison where AgeDistance=0 (sample P816_B8320 and M148_X5461). I'm not sure this is the best way to present the data. We already know from previous research that the microbiome of adults and young juveniles should look different and that IgA/mucin is associated with age in hyenas. I'd prefer a visualization of your data than model predictions. I think it would be more effective to stratify by age groups and look for associations within each. Perhaps just ordinate the microbial diversity for juveniles, see which axis separates the data by f-IgA and plot the axis loadings over f-IgA (then repeat for adults).

>> The reviewer interpretation is correct: those are predictions. We have redesigned these panels, now in revised figure 3B. This panel shows the predicted effect of IgA/mucin distances on microbiome similarity controlling for all other variables, and helps visualise the effect on the microbiome similarity of the interaction between age and immune measures. To address the

concern that sample sizes vary across age differences, we have revised the figure. Instead of showing this predicted effect at e.g. age distances of 0 and 1, we show the predicted effect of each immune measure at the mean age differences and at +/- 1 standard deviation, reflecting the true distribution of the data.

We appreciate the suggestion to replace this figure, the problem with the suggested approach is that it does not fully account for other covariates and the interpretation is not, in our view, improve substantially (e.g. IgA increases with PCoA1 loadings vs IgA differences decrease microbiome similarity). We added the suggested ordination figure to the supplementary materials (figure S4), with A) all samples (both adults and juveniles), B) adults only and C) juveniles. We have also added the vector of the correlation of the continuous host and environmental variables.

R1C30 Longitudinal analyses: I found it odd that even though temporal distance (time) was found to be significantly influential for all microbial groups tested, this data wasn't visualized. Was seasonality explored at all? I'd expect it might affect parasite load. I'd love to see visualizations showing microbial diversity metrics (α diversity and beta diversity ordination loadings), the abundance of microbial groups, and immune measure levels over age AND over time.

>> Revised figure 2 shows the effect of temporal distances in relation to all other variables. We have also added season to all models, and this effect can be visualised in revised figure 2. We have created ordination plots (see answer R1C29 on age ordination), that also includes samples coloured by year of sampling, figure S4.

R1C31 How were longitudinal samples handled? Including multiple samples from the same animals could inflate the association between the microbiome and immune measures. These longitudinal samples could also be used to analyze within individual changes over time (especially the individuals with 3 samples) although this may be outside the scope of this analysis.

>> We accounted for repeated samples from the same individuals by using a multi-membership random-effects framework that accounts for individuals in each pairwise comparison, i.e. the animal ID of each sample pair is included as random effect (animal A, animal B). We clarified this by rephrasing this sentence: "We used a multi-membership random-effects framework that accounts for individuals in each pairwise comparison (e.g., Individual A, Individual B)." lines 289-291

R1C32 FIGURE 4: See above for comments related to the host & unclassified DNA...

>> We hope that our new formulation now clarifies this aspect. See reply to R1C6 and R1C7.

R1C33 4A,C: I don't find these graphs very interesting aside from the R2 value, they could be supplemental. What is the shading? Are these credible or confidence intervals?

>> We have removed these panels from the figure.

R1C34 4B,D: These plots are more interesting but don't give information on directionality. I'd suggest that rather than sorting by importance, you provide a heatmap of samples sorted by f-IgA or mucin measurement levels and the relative abundance of these taxa.

>> We agree that giving information about directionality is important. We provide partial dependence plots for each of the 20 most predictive taxa of f-IgA and f-mucin, figure S5 and S6. These figures show the marginal effect each taxa has on the predicted outcome of the random forest models. Unsurprisingly we find many non-linear effects. We added heatmaps of samples as suggested, figure 4C. We sorted by f-IgA and f-mucin feature importances to be consistent with figure 4A and B. We chose not to sort by relative abundance of these taxa as we don't have absolute abundances and we keep our interpretation on abundances to comparison between samples and not between taxa, e.g. taxa A is more abundant than taxa B. See reply to R1C11 and supplements note "**Interpretation of relative abundances**". Note that we chose to keep the plots showing the cASVs sorted by feature importance as we think that they deliver an important and clear message as well.

R1C35 4E: I like the use of a network but would like to see it without host DNA. Also, please keep the colors chosen for your microbe groups consistent throughout the figures.

>> We adapted the figure to keep the color code for groups consistent throughout the figures. The new random forest models include, in addition to the relative abundances of taxa, host and environmental predictors too, and interestingly host DNA and prey DNA are no longer among the 20 most variables predicting immune measures and are likely under the influence of host and environmental effects. The revised network includes all taxa and the highlighted "important" taxa are depicted.

Minor comments:

R1C36- Title: "Symbiont" doesn't seem appropriate here, especially since the data selection for the eukaryotes was geared towards parasites. Also be careful of implying causation.

>> Thank you for this comment. We have changed the title to remove the implication of causation. We would like to clarify that we used the term symbionts throughout this study based on the definition the coined by de Bary (1879) in "Die Erscheinung der Symbiose": "the living together of differently named organisms", "the living together of organisms of different species that are associated with parasitism, mutualism, etc." (DOI 10.1007/s13199-016-0409-8). Since this term is not used consistently throughout the microbiome, ecology and evolutionary biology field, we have decided to remove it completely from our manuscript.

R1C37- Line 57: "Intestinal mucosa" often refers to the GI layer (i.e., the epithelium, lamina propria & muscularis). Did you mean the "intestinal mucosal epithelium" of the small intestines/colon (remember that distal GI tract regions have stratified epithelium). Keep in mind which regions fecal samples likely represent.

>> Thank you for this input. We corrected it accordingly. Lines 57 - "The intestinal epithelium, a complex layered set of cells covered by a mucus layer, is the first line of defence...".

R1C38- Line 61: Recommend “can be associated” since perturbations can also lead to a new baseline community that is not necessarily unhealthy (e.g., a diet shift)

>> Corrected accordingly, to read as “Perturbations of this community can be associated with intestinal diseases, often accompanied by increased mucosal permeability and disrupted immune responses” lines 61-62

R1C39- Line 78: “inter-kingdom”?

>> We rephrased accordingly, lines 79-81 “Studies on immune responses to parasites (i.e. protozoans and helminths) and fungi are often performed in isolation from the rest of the intestinal community, although interactions with other taxa likely shape both the gut microbiome and immune responses”.

R1C40- Line 83: Over-generalization. This sentence refers to “wild animals” in captivity- in the context of this study, readers may assume “captivity” means zoos. However, two referenced studies (27, 29) are on mice (I’d argue lab mice have unique husbandry) and the other sampled humans (28). The latter included published data on wild/captive mammals, yet wasn’t designed to explore captivity. Ley found that host phylogeny/diet overshadowed other covariates. More recent studies of wild mammal bacteriomes suggest that certain host clades (e.g., Perissodactyls) are resistant to the captivity-induced changes (see doi: 10.3389/fevo.2021.785089 and DOI: 10.1093/icb/icx090). Suggest rewording and/or adjusting references.

>> The sentence the reviewer is referring to is “Wild animals living in natural environments have a distinct microbiome compared to their captive counterparts, characterized by a higher diversity of eukaryotes”. We opted for the adjustment of our references, thank you for noticing this. Added references: captivity effects 10.1038/s41598-019-43875-3, 10.1038/s41467-021-25732-y, 10.3389/fmicb.2022.832410, 10.3390/ani13101625 , 10.3390/microorganisms12071419 and diversity 10.3389/fmicb.2021.665853 and 10.1007/s13199-022-00853-0

R1C41 - Line 87: Flow is a bit confusing. Line 80 gives a nice transition to wild mammals, but then 83 refers to mice/humans and then 87 is vague (yet the refs have transitioned to studies of wild animals). IMO, this paragraph should focus on the growing knowledge surrounding wild mammalian population microbiomes, the feasibility of non-invasive sampling and the implications for conservation or understanding microbiome-environmental interactions.

>> As we have adjusted our references in response to your previous comment, the whole paragraph is now focused on mammalian microbiome in wildlife.

R1C42- Lines 106-111: Really nice job stating the hypothesis and tying in the need for mixed community microbiome analysis using refs to previous microbiome work and the Anycyclostoma/mortality example! I would try to flesh out the sentence about the previous microbiome, parasite and immune measure studies that were recently done (I believe by

your lab) to make it clear what findings informed your current work and how this study helps complete the story.

>> Thank you for your positive feedback. We suspect the reviewer is referring to the following sentence: “Previous research has shown that host characteristics (e.g. age) and environment (e.g. clan membership) affect parasite infections and intestinal bacterial composition . Furthermore, faecal IgA and mucin reflect *Ancylostoma* load, a parasite shown to reduce longevity in the Serengeti population ”. We rephrased it to provide more details and it now reads as follows: “Previous research on Serengeti hyenas has shown that host characteristics such as age and the social environment, including clan identity and structure, affect parasite infections ^{57,61,62} and gut bacterial composition ^{35,63,64}. Age is a particularly important factor, as it affects parasite composition ^{57,61,62}, gut bacterial composition and modulates faecal IgA and mucin⁶⁵. Furthermore, faecal IgA and mucin reflect *Ancylostoma* faecal egg load, a parasite shown to reduce juvenile survival and longevity in the Serengeti population ^{57,65}.”. Lines 113-118

R1C43- Lines 115-125: Impressive dataset and I appreciate the use of host genotyping. Is this the only sentence referring to the host genotyping methods? The reference is good, but you might want to say whether this leveraged an existing hyena reference database (and if so, mention the database coverage in terms of the current population). Perhaps mention in the supplemental methods.

>> We suspect that the reviewer is referring to the sentence “Maternal identity was determined based on nursing observations at the communal den(s) and was further confirmed by DNA microsatellite loci analysis” we added “Maternal identity was determined based on nursing observations at the communal den(s) and was further confirmed by DNA microsatellite loci analysis ⁶⁹. Approximately 1250 hyenas were genotyped at 9 microsatellite loci (see ⁷⁰) representing about 41% of the individuals born into one of the three study clans since the start of the project in 1987.” lines 150-152.

R1C44- Lines 126 (and 241 etc...): In general, figures presenting actual data should be called out in the results.

>> We agree and have removed references to figures in the Methods.

R1C45- Line 138: I'd expand on the sample collection/field conditions. How were the samples collected immediately (without risk to researchers)? How was anthropogenic manipulation (Line 95) minimized in this study? Were cool packs in a cooler? Approximately how long did samples sit before reaching the field station?

>> We further expand that section on lines 166-174 “Non-invasive faecal samples were opportunistically and immediately collected after defecation. The researchers used the same research vehicle to which animals are habituated and were careful not to be seen when collecting the samples, by positioning the vehicle very close to the faeces, to guarantee the safety of the researchers and minimise disturbance to the animals. Samples were collected in individual labelled bags and refrigerated in cool boxes with frozen ice packs in the field until transport to the field station (no more than 3-4 hours later). At the field station, they were

mechanically mixed and aliquots stored at -10°C until their transport to storage at - 80°C at the IZW ⁶¹. “

R1C46- Line 148: Suggest “Microbial DNA extraction” to differentiate from the host genotyping?

>> The reviewer is referring to the name of the section “DNA extraction”. We have changed the subsection to “**DNA extraction of faecal samples for microbiome assessment**”, line 183

R1C47- Line 149: Perhaps specify what the NucleoSpin Soil kit targets. I was unfamiliar and the Takara blurb only mentions bacteria, yeast, and fungi. Maybe include a reference for its use in helminths (e.g., doi: 10.1186/s13028-022-00624-3)

>> We have added that we have previously assessed parasites with this kit. “The NucleoSpin®Soil kit has previously been successful in uncovering bacteria and eukaryotes, including the parasitic phyla apicomplexa, nematoda and platyhelminthes, in faecal samples from spotted hyenas³⁵” lines 194-196

R1C48- Line 215: I interpret “repeatability” to mean the comparison of one sample repeatedly sequenced under equivalent conditions (technical replicates, e.g., PCR duplicates). Are you referring to technical replicates? It seems more likely that you are comparing longitudinal samples from hyenas that were sampled over time. This should be clarified given that these are very different research questions.

>> The reviewer refers to the sentence “We tested the effect of individual repeatability on β -diversity measures in 78 samples from 37 individuals in the overall microbiome...”. Thank you for pointing out another important case where the same word can have different meanings. In this case we are referring to the individuality of microbiome phenotypes which can be estimated via their repeatability through time. We are using this word from the classical definition: the proportion of total phenotypic variation that is attributed to individuals. We clarified it now in the text accordingly, see lines 267-268 methods “We tested the effect of individuality of microbiome phenotypes (estimated as repeatability through time)...” and in the results “The intra-individual microbial difference was low or nil for the overall intestinal microbiome (dICC=0.095, SE= 0.033),...” lines 374.

R1C49- Lines 221-222: I interpret “dyadic” to mean two linked individuals, and in this case, one could take it to mean mother-pup dyads. Do you mean “pairwise” here? State the specific beta diversity measure used (instead of “(distances)”). I’d reword or remove “(excluding within-sample comparisons)” since for a moment I thought you were referring to alpha diversity (but I think you mean comparing a sample to itself which would have BC=0)?

>> The reviewer is referring to the sentence starting with “We tested for the association of immune measures while accounting for the effects of other known or expected host, social, and ecological variables on the β -diversity of the intestinal microbiome composition using dyadic comparisons (distances)...”. To clarify this aspect we rephrased as suggested: “We tested for the association between immune measures and the β -diversity of the intestinal

microbiome composition (microbiome similarity among compared samples), while accounting for the effects of other known or expected host, social, and ecological variables, using pairwise comparisons (distances) among samples (see lines 276-277). We have used dyadic and pairwise comparisons interchangeably throughout the manuscript. In the revised version we have avoided using the term dyadic, unless when referring to observations between 2 individuals.

R1C50- Line 228: Will all readers be familiar with Stan? Perhaps say “in the Stan probabilistic programming language” and list the R interface used (RStan, cmdstanr?) + version

>> We rephrased this sentence as suggested: “implemented in Stan probabilistic programming language through the brms R package v. 2.19.0⁹¹” see lines 288-291.

R1C51- Line 232: Looks like “the Sampler” should be lower-case when used as such https://cran.r-project.org/web/packages/brms/vignettes/brms_overview.pdf

>> Thank you we replaced it by “the sampler” see line 293.

R1C52- Line 244: Did you test for all interactions in the model before arriving at the final version? The reference you provided did find interactions between age and immune measures, but since here you are testing for influence of immune measures/environmental/social covariates on microbial diversity, you may reveal different interactions. Glancing at the code it appears that the immune/age interactions were manually entered but I don’t see that the other variables were tested for interactions (i.e., separated via “+” rather than “*”).

>> That is correct, we included only two interactions based on previous knowledge, including our own previous study, that is we did not test for all potential interactions, only the hypothesised one. We have slightly edited our previous sentence, to read “We included two interactions: one between age and f-IgA and one between age and f-mucin, based on previous findings⁶⁵” lines 283-285.

R1C53- Lines 248-250: Recommend shortening this section by adding `splitrule=variance` and `min.node.size=10` to the previous sentence (since they were the same for both immune measures) and then just specify the `mtry` for each

>> Edited as suggested lines 316-318- “The final values used for the models were `splitrule=variance` and `min.node.size=10` and `mtry = 55` for the one predicting f-IgA, 266 for the f-mucin.”

R1C54- Line 253: Describing taxa importance as permutation importance is a bit redundant and doesn’t tell me much. Perhaps specify that this is a command argument (`importance=“permutation”`) and explain what permutation importance means (something like “a variable’s influence on the model’s predictive performance”)

>> We rephrased the sentence as suggested “The importance of each variable was assessed based on the command argument `importance = “permutation”`. This argument provides a variable’s influence on the model’s predictive performance. lines 320-322

R1C55- Line 253: Give the mean and range of samples collected for those 37 hyenas

>> We added this information lines 269 “sampled 2 to 3 times, with a mean sampling per individual of 2”.

R1C56- Lines 298-300: Contamination/final dataset. I realized contamination should be minimized due to the microfluidics approach but what was the percent of reads/ASVs removed by Decontam? Comment briefly on the ASV/sample drop out post-QC. On that note, 301 bacteria + 691 eukaryotes= 992 cASVs (not 999). Among eukaryotes 420 were fungi and 25 were known pathogens, what about the other 246 cASVs?

>> Thank you for bringing this to our attention. There were 34 ASVs out of 21658 ASVs that were removed using the decontam approach. This has been added to the text, lines 223-226 “34 contaminant ASVs with decontam v. 1.21.0⁷⁶ using “prevalence” and “frequency” methods (method = “combined”) “ and figure S1. Figure S1 also shows the sample and ASV drop out throughout the bioinformatic pipeline. We had 7 cASVs that were unknown at the domain level. Among eukaryotes, 464 cASVs were neither fungi nor considered parasites. Figure 1 describes the phyla in this study.

Reviewer #2 (Remarks to the Author):

In their introduction, the authors provide sufficient background regarding the state of research on the microbiome and lay out a compelling rationale for the study of wild animal populations as it relates to questions of host immunity and microbiome interactions. They proceed to analyze a longitudinal dataset of hyena samples to understand how host immune markers (f-IgA and f-mucin) relate to different microbial profiles, suggesting that microbiome composition is more similar in pairs of individuals with similar immune markers or age. Some microbes were further identified to be predictive of host immune markers, suggestive of stronger host-microbe interactions. Overall, the authors have conducted a novel study that contributes to the field; importantly, it looks at the microbiome more holistically by taking into account bacterial and non-bacterial members of the microbiome, and further does so in a wild population where multiple sources of natural variation influence the already complex interactions between host immunity and the microbiome. There are areas for improvement, specifically in considering the interactions between microbial groups and how those interactions may jointly influence immune markers or mask one another's effect on those markers. Addressing this area will strongly contribute to the importance of this work and further supplement the authors' argument regarding microbial cross-talk and the importance of a holistic approach to microbiome research. Some comments for the authors' consideration follow, with the aim of strengthening the overall presentation of the data and the analyses in this manuscript.

>> Thank you for your interest in our study and your very helpful comments. We appreciate that you found the questions and population we investigate interesting. We addressed your suggestions and think that they have indeed helped us clarify several aspects related to the presentation of the data and analyses.

R2C1 In lines 190-199, the authors describe their screening methodology for preparing sequence data for downstream analysis. While some reads discarded in this manner may not align or have biological relevance, it may provide a stronger argument to attempt an alignment of these ASVs to database(s) (i.e. SILVA 138.1) to check for potential matches and then provide further rationale for discarding these sequences. For example, some sequences may represent taxa that seem biologically unlikely to be incorporated into the hyena microbiome and should thus be discarded. As it currently stands, there is a likelihood that some rare taxa are not incorporated into the downstream dataset, which could disproportionately affect certain beta-diversity measures used in the analyses (Jaccard distances, as one example). The current approach biases the initial dataset towards homogenization rather than reflecting the desired natural variation that the authors initially note as a desirable feature of studying wild populations.

>> We agree with your point of view and acknowledge that we are removing likely errors with the cost of losing rare taxa. We have changed the filter parameters and now only remove the singletons (taxa present in only one sample). Figure S1 shows the bioinformatic pipeline. In the new version of the manuscript out of 21658 ASVs, 4706 ASVs were retained after the quality filtering (removing singletons and samples with fewer than 100 reads). The taxa retained in this more broad filter are mostly eukaryotes (previously there were 301 cASVs from bacteria and 691 cASVs from eukarya, now there is 407 cASVs from bacteria and 1183 cASVs from eukarya). We further include in the supplements a figure (figure S2) that shows the phyla of all removed singletons. We rewrote the corresponding sentences in the methods and rerun all models “We further removed amplicon sequence variants (ASVs) present in only one sample and samples with less than 100 reads.” lines 226-227

R2C2 In lines 232-236, the authors describe testing the effect of individual repeatability on beta-diversity measures in a subset of samples (78/210) from a subset of individuals (37/165). It would be helpful to understand how these subsets were selected. Further, each individual is presumably duplicated in this subset of samples, but are some individuals replicated in triplicate given the total number of samples in the subset? Overall, it may be helpful to include a supplementary metadata table that clarifies the relationship between the 210 samples and 165 individuals, including frequency of repeat sampling for individuals particularly as it relates to this testing of repeatability on beta-diversity measures.

>> We agree that we had not put much effort in describing our selection of samples in the original manuscript. Figure S1 clarifies the initial sample size and samples removed. Figure 1A shows more clearly shows the individuals with repeated samples and we include in the text the range of repeated samples lines 268-269 “78 samples from 37 individuals sampled 2 to 3 times, mean sampling per individual of 2. ”. In the methods, we clarified that all samples were selected opportunistically whenever a known individual from the study clan is observed defecating. Due to the opportunistic nature of our sampling strategy there can be very large variation in the number of samples collected per individual “Non-invasive faecal samples were opportunistically and immediately collected after defecation. The researchers used the same research vehicle to which animals are habituated and were careful not to be seen when collecting the samples, by positioning the vehicle very close to the faeces, to guarantee the safety of the researchers and minimise disturbance to the animals. “lines 166-174.

R2C3 In line 239, the authors reference dyadic comparisons across samples, excluding within-sample comparisons. The number of dyads (19701) seems appropriate for 210 samples, but this cannot be validated without further detail on the nature of the sample collection, number of repeat samples per individual, etc. as described in the comment above.

>> We agree and hope that our additions, detailed in response to your previous comment, allow us to provide a more detailed background about our strategy regarding the sample collection and sample selection, including an overview of repeated samples for each individual.

R2C4 Line 261 references Table S1, which includes two additional beta diversity measures that were used to analyze the effect of different factors as it related to microbial composition across different taxonomic groups. Table 1 in the paper includes a similar analysis focusing on Bray-Curtis distances. For ease of comparison, it would be helpful to include the BC-distance analysis in Table S1 as well alongside the other two metrics. Further, there is no discussion of these results from the other distance metrics and how they support or diverge from the BC-distances; they seem to largely be consistent. By addressing all three and their similarity and dissimilarity, the authors can provide a more nuanced argument for the observed associations.

>> Thank you for bringing this up. As suggested, we included the BC-distance analysis in Table S1 to facilitate the comparison. We include the rationale of adding other β -diversity measures in the methods and indicate the consistency of our finding among different measures, "To verify the robustness of our results, we repeated all models using both occurrence, i.e. presence/absence (Jaccard distances) and abundance-based β -diversity measures (Aitchison distances and Bray-Curtis distances), see Table S1 - section supplementary tables, supplements file. The results are largely congruent as indicated by the results in table S1, so we decided to report our results based on the Bray-Curtis distances." lines 298-302 and "We evaluated the effect of various host and environmental factors on the overall microbiome membership (Jaccard distances) and structure (Aitchison and Bray-Curtis distances). We observed consistent effects for all β -diversity metrics (Table S1), we, hereafter, only report the results on community structure based on Bray-Curtis distances" lines 389-392.

R2C5 In lines 279-281, the authors reference using BLAST to "confirm" the taxonomy of ASVs of interest. However, in the provided table (S3), only 8/33 taxa are matched at 100% alignment, most to "uncultured bacterium" rather than a genus or other taxonomic identifier, which was the purpose of the BLAST. Softer language or a caveat regarding this confirmation may be more appropriate.

>> We agree and rephrase this section in the methods to read as "As many taxa were not classified at phylum or even domain levels, we attempted to minimise uncertainty arising from this by inspecting the taxonomy of these taxa through searching for the most similar nucleotide sequences using NCBI BLAST searches⁹⁵ and inspecting the top hits and corresponding sample source." see lines 325-330

R2C6 In lines 282-288, the authors describe a co-abundance network analysis identify associations between potentially ecologically relevant taxa. Does using all of the data (without the filtering down to 20% prevalence or higher) affect the results? Given the prior

filtering step to remove artifacts and rare sequences, further filtering seems unnecessary. Sparsity of sequences does not inherently obviate ecological relevance, so relevant data may be excluded by this filtering - especially as this seems to have dropped the number of cASVs from 999 to 143. It would be helpful to either see a comparison of the analysis, with and without filtering, or for additional reasons to be provided for filtering down to 20% prevalence and higher.

>> Thank you for this suggestion. We agree with the reviewer and now include a network without any previous filtering, figure 4D.

R2C7 In line 292, the authors reference sampling the intestinal community of 158 individuals; this seems to contradict the 165 referenced in lines 125-126. This 158 number is also referenced in the abstract in line 36, in the Figure 1 caption in line 310, and in the Figure 2 caption in line 337.

>> Thank you for spotting these potentially confusing numbers. We hope the overview of figure S1 now clarifies the differences in sample size.

R2C8 In discussing the association between host immune measures and microbial groups, the authors should consider controlling for similarity or dissimilarity of other microbial groups, which may more accurately attempt to capture the "cross-talk" between communities. As an example, the authors note alignment between bacterial and parasitic profiles (lines 419-420). Therefore, the analyses where the effect of a particular community (i.e. bacteria) on f-IgA or f-mucin are likely not entirely isolating that community; there is an entanglement between the bacterial and parasitic profiles, in this example, that may be falsely attributed to one or the other or resulting from some interaction between them. The authors might consider the following:

1. Noting the potential effects of this interaction in the results, which would require less effort.
2. Attempting to measure this interaction and including it in the model, which would require more effort but strengthen the manuscript and arguments therein. The same could be repeated for each pair-wise community interaction, as relevant.
3. Control for the interaction such that the resulting data only reflect, to the extent possible, the influence of just one microbial community, which would still require some effort.

This expanded analysis (options 2 or 3) would further strengthen the discussion on the co-abundance of these microbial groups.

>> Thank you for this comment. We agree that the interactions between microbial groups may jointly influence the immune measures that we investigate. This is a very interesting aspect and also hard to disentangle these effects, given the complexity of the gut ecosystem. Previously we modeled the host and environmental effects on the decomposed group similarity (parasites, bacteria and fungi) with a multivariate multilevel model. This indicated that parasites, bacteria and fungi similarities were correlated. In the revised version, we include group similarity as a fixed effect. This is a first step towards accounting for interactions among taxa. We further explore how individual taxa and host and environmental factors are associated with immune measures. To do so, in the random forest regression models (predicting immune measures), we now include host and environmental variables together

with taxa abundances. Random forest regressions do not assume independence and linearity of the predictors, which is an advantage when exploring the effects of interacting taxa and interacting host-microbiome. We acknowledge that further research is needed to fully understand the nature and mechanisms of this interaction in the discussion, e.g. “Because we measured broad-acting measures of mucosal immunity, we cannot infer specific mechanisms of interactions between the different members of the gut microbiome and the host. In future, measuring antigen-specific IgA isoforms and mucin types together with assessments of intestinal communities may shed light on the finer details of the regulation of individual taxa in the biomes.” lines 605-607. We have updated the methods and result section to reflect these interactions and updated revised table S2 with model parameters and figures 4A.

R2C9 While age, time, and parentage are all considered in the evaluation of the similarity of both host immune markers and microbial profiles, sex seems to be a known data element (except for those individuals who were too young) that is not addressed for its potential impact or lack thereof. This would likely only require a sentence to address and minimal effort if the data is already part of the existing metadata for these samples.

>> Thank you for the suggestion. As detailed in our response to a comment made by the other reviewer (see response to R1C23), we have now conducted a novel analysis that specifically aims to test for a sex effect among juveniles (as our samples in adults are restricted to females see our response to R1C14). Table S3.

R2C10 Further, is there any existing data from the long-term study of these individuals that would allow for a discussion of the impact of environmental effects, seasonal effects, or dietary effects? Including these data may be significantly more effort, but would complement existing discussions and observations of these trends in the literature and add to the comprehensiveness of the analysis in this manuscript.

>> As suggested, we added season as a new covariate in all models, to cover a potential additional effect of environmental effects, see figure 2, table S1, S2 and S3. We would argue that using the date of collection and clan memberships should also capture some environmental variation.

R2C11 Finally, is there existing data on the social interactions between individuals? Multiple publications already point to the effect of social interactions on microbial similarity, as the authors themselves cite, but there are novel data here regarding other microbial groups. A social network analysis, even with a subset of data, might further complement the existing findings if feasible. This would require significant effort, assuming the data exists but needs to be manipulated into the correct form for analysis.

>> This is a very exciting field of research and we would love to be able to conduct a social network analysis to investigate the relationships between social interactions and the microbiome. Unfortunately our behavioural data have not yet been prepared to construct social networks and the large temporal scale of our study means that a large proportion of individuals included here do not overlap at the time of sampling; this would thus be a project on its own requiring the analysis of new samples collected within a relatively short period of time.

Revision: #3 28 March 2025

Dear Editor(s),

We thank you for all the comments and for the opportunity to publish a revised version of our manuscript. Below are the answers to all of the reviewer's new questions. All line numbers refer to the revised manuscript.

We hope that we addressed these final concerns and that you will find this revised manuscript suitable for publication in *Communications Biology*.

Sincerely,
The authors

REVIEWERS' COMMENTS:

Reviewer #2 (Remarks to the Author):

The authors have made significant efforts to revise the manuscript as per reviewer feedback. Overall, the authors present a much stronger argument on the relationships between mucosal immunity and the gut microbiome, including both bacteria and eukaryotes. Changes to figures, explanation of sampling and the more detailed methods, and the added nuance throughout the text are notable edits. The authors are careful not to overstate their results, particularly given challenges in interpreting fecal-associated markers and microbiome data with well-characterized host-immune interactions in the small intestine. The authors' approach to labeling edits is also appreciated. There are some additional considerations, provided below, the authors might consider to further evaluate the manuscript prior to publication; these are largely minor as the authors have submitted an excellent manuscript for consideration.

>> Thank you very much for carefully reviewing our manuscript and your additional suggestions. We greatly appreciate your recognition of our efforts and for considering our revised manuscript excellent. We couldn't have reached that state without all of your inputs and suggestions.

Given the importance of age on beta-diversity (F2) and f-IgA and f-mucin (F4), is it possible to tease out how much of that effect derives from comparisons between adult-juvenile pairs vs. other pairs? It would be important to highlight if there was a non-adult-juvenile age-related pairwise comparison that was driving these trends. Given the longitudinal data, are there any samples from juveniles who grew into adults across samplings that could be used as another subset for further exploring these trends? Similar to data in Figure S3, if there is information on age of sampled individuals, that may also be an informative histogram.

>>This is an interesting point and suggestion. Age has a considerable influence on immune measures in spotted hyenas, particularly before adulthood and during the transition to adulthood. In our analysis focusing solely on juvenile pairs, we find a similar effect of immune measures to that observed in models considering all pairs. Our analysis does not have sufficient sampling resolution to study in detail the effects of immune measures in adulthood, including possible interactions between ageing and this effect. We highlight this point in the new version of the discussion, lines 567-569. The proposed

longitudinal analysis is unfortunately not possible because the “juvenile to adult” subset has a small sample size (22 cases). Figure 1A shows the detailed age at sampling, including longitudinal samples (from the same individual).

The new supplemental Figure S1 is generally clear in describing the methodological pipeline, removal and filtering of samples, and combination approach. Further, the primer table is a clear and useful resource to clarify the approach for the study design. The authors’ broad targeting methodology for amplification and sequencing can result in multiple sequences from a single taxon, which is well-acknowledged. In Figure S1, the authors state “After combining ASVs likely produced from the same taxon,...”, similar to lines 246-249 in the main manuscript. It is ambiguous if this combination is the step resulting in the cASVs (when multiple ASVs are included) or a separate step. If this is not a separate step, there may be concerns about how some measures, like Bray-Curtis distances, are calculated, as these measures rely on abundance counts, which could result in a bias towards taxa sequenced multiple times under different primers. In S1B and/or Table 1, it may be helpful to clarify the process for avoiding duplicate counts of taxa sequenced multiple times given the sequencing approach when creating cASVs and when calculating Bray-Curtis distance metrics. Given the general similarity of observed results with other beta-diversity metrics, this is likely a minor concern.

>> Thank you for this suggestion to make our statements clearer. We merged the ASVs into cASV that are likely from the same taxon, to avoid duplicate counts. We agree there might be a bias resulting from this approach, but we are also confident that it does not substantially affect the biological results, based on previous work in which we had access to both single and multi-amplicon sequencing data (Ferreira et al, 2023 and 2024). We also report the results using Jaccard distances β -diversity that do not rely on relative abundances.

The purpose of Figure S2 seems to be a general indication of removed phyla due to filtering, but it is only limited to those taxa that appeared as singletons. Is there a reason to not include those taxa that were filtered for having less than 100 reads? This is addressed in the figure caption, but apparently not included in the figure itself.

>> Figure S2 includes the phyla of all singletons present in all samples, to provide rationale for discarding them, as a reply to previous R2C1. We consider singletons as likely sequencing errors. We did not include in the figure the ASVs discarded in the samples that had less than 100 reads, as these are discarded not because of their likelihood to be errors, but rather because the samples are not sequenced deeply enough to represent the taxa in it. In the figure legend we wrote the final number of the ASVs kept, which is indeed confusing since the figure shows the excluded ASVs. We removed this from the figure legend.

Text edits:

Simple to single, plural ASVs (Line 257) - “For simplification purposes, we use the term cASV regardless whether it refers to a single or combined ASVs.”

>> Corrected in the text in accordance with the suggestion.

E to D (Line 504) - “... cASVs for predicting f-IgA and f-mucin. D) Top: Co-occurrence network of 1597 taxa...”

>> Corrected in the legend in accordance with the suggestion.

Journal: Communications Biology

Manuscript #: COMMSBIO-24-4557-T

Title: Spotted hyena gut cross-talks: Symbionts modulate mucosal immunity

Current revision: v0

Due: Sept 4th, 2024

Manuscript summary: In this study, the authors investigated immune measures (IgA and mucin) and used a multi-amplicon approach to profile bacterial, fungal and eukaryotic parasitic communities in 199 fecal samples collected from 158 wild hyenas over a ~14 year period. This work follows up previous studies that demonstrated individualized microbiomes in this population (influenced by factors such as age and prey availability), and a positive correlation between fecal IgA and mucin concentrations and *Ancylostoma* egg burden (also correlated with age and host survival). The goal of the current manuscript was to determine the relationship between immune measures and gut microbial community composition. The hypothesis was that host-associated microbial community composition was strongly associated with fecal IgA and mucin levels. The authors used probabilistic models and machine learning in their analysis. Their main findings were: 1) immune measures predicted microbiome similarity in an age-dependent manner; 2) immune measures were more strongly correlated with the bacteriome than parasites and even more so than fungi and 3) microbiome composition could be used to predict immune levels. Specific taxa driving this association were identified, including *Ancylostoma*.

Overall impression of the work: This study leveraged a dataset that was impressive both in terms of sample size and study duration. Given the recent research coming from the same group of researchers that has shown this hyena population has individualized fecal microbiotas and fecal IgA/mucins levels/parasite burdens associated with survivorship, it seemed fitting that this study bridge these factors by linking IgA/mucin to microbial diversity. That said, I found it hard to keep track of exactly which taxa were being targeted, especially because certain results highlighted taxa that I had assumed were off-target (vertebrate DNA and organisms unclassified at the level of domain/phyla). Admittedly, I have not worked with multi-amplicon microfluidic PCR myself, but without a clear description of data normalization and consistent data subsetting, I was not convinced that the relative abundance of certain microbial groups was directly comparable. I also felt that the bulk of the figures revolved around the Bayesian probabilistic model and pairwise comparisons of both diversity measures and covariate measures, which are not intuitive for the reader. I believe that the incorporation of more straightforward visualizations along with such analyses will be more effective. Overall, I think this manuscript has great potential but will require some work to clarify the methods and better visualize the results.

COMMENTS TO THE AUTHORS:

Major comments:

Data reproducibility & confusion about methods: The following issues made it difficult for me to fully understand and evaluate what was done in this analysis.

- **Code:** Thank you for providing code, but it looks like `rFit_IgA.rds` is missing from the `tmp` folder so I was unable to reproduce the latter part of the analysis. I'm also unable to link the sequences to the ASV IDs used in the manuscript when starting with the `fPMS.rds` phyloseq object. (Minor note- for the next submission, please organize the code according to the figure numbers so chunks are easier to locate. For example, the code refers to "Fig5" but there is no Fig 5.)
- **Primer selection:** I'm confused about which taxa were targeted in this study. The primer table is missing some information and some primers aren't mentioned in the manuscript (i.e., CytB, HSP70, tRNA and COI). Were all these primers used in this study? Was qPCR done? Was host DNA targeted in addition to microbial? The manuscript divides the data into 3 groups: bacteria, fungi & parasites, so I would tidy the primer table and organize it as such. In addition to the broad "Target" column, add detail what was targeted by specific primers at a finer level (e.g., protozoa/helminths/fungi within Eukaryota). Add citations to back up primer choices. This should be provided as a supplemental table/file and I would consider adding a simplified table to the main manuscript. Below are some related comments regarding the associated text that may clear up confusion:
 - Lines 187-191 are redundant. Remove the first sentence and clarify the targets of each (e.g., "the UNITE database for fungal ITS...").
 - Line 205 refers to "four groups of cASVs" but only three are listed. Plus, are they all really "cASVs"? For example, only one primer pair was used to target 28S, so wouldn't these sequences just be "ASVs"? If both cASVs and ASVs were used this should also be specified in the primer table.
 - Lines 205-211: Just list the three broad categories (bacteria, fungi & parasites). Why are specific phyla are listed here? If these were the targets, include that information in the primer table. If these are lists of the successfully sequenced phyla, this information should be in the results.

Non-microbial reads and unclassified taxa: I'm not sure why the authors decided to retain vertebrate DNA in the analysis since the focus was on the microbiome. They also retained unclassified ASVs. Since these are typically filtered out early in the analysis, I suspect removing these ASVs may significantly affect the results.

- **Off-target amplification:** Was the hyena host DNA specifically targeted here? Or was this host contamination as is common with certain primers? Most microbiome studies aim to avoid amplifying host DNA so that a larger proportion of the reads are microbial targets. Some even use PCR blockers to avoid host contamination. If the host DNA was of interest, wouldn't *quantification* be more important (i.e., qPCR, ddPCR, WGS)? If the goal of using host DNA was to examine inflammation, wouldn't immune assays or even stool blood content be more appropriate?

Furthermore, there I don't see a "Crocota" designation in the tax table and the only cASV from Vertebrata. This BLASTs to many mammalian species (e.g., 1st hit is an Etruscan shrew). How is it known that this is hyena DNA and not prey DNA? What is more confusing is that at line 486 in the discussion the authors state "our results indicate that prey DNA is negatively correlated with host DNA". I cannot find any analysis to support this claim. How could this comparison be made with only one cASV corresponding to Vertebrata? Several cASVs correspond to arthropods, but wild hyenas must be eating vertebrates as well.

- **Unclassified taxa:** What was the rationalization behind keep taxa with unclassified kingdoms and domains? Typically, taxa that aren't assigned to the level of phyla are removed prior to the data analysis since they may be spurious. This is especially curious considering many of these taxa were identified in the machine learning model. I appreciate that the BLAST table was provided, but the fact that so many hits were to unclassified bacteria or even "unclassified organism" makes me think they should have been filtered.

Merging ASVs: Line 201 refers to the Ferreira et al., 2023 manuscript for the detailed methods of merging ASVs. However, it appears that in that study the tree was trimmed down to just *Eimeria*. The current analysis seems more complicated since multiple primers were used on multiple targets. Perhaps another reference is missing from the claim in line 202 "is now extended to all annotated genera" that contains such a workflow? If not, I recommend rephrasing that to something like "and is extended to multiple microbial targets in the current study" and expanding on the workflow description (e.g., including supplemental trees, perhaps before and after merging into cASVs).

Normalization: How was this data normalized? The all-in-one approach of using the Fluidigm Assay has obvious benefits, but one would assume that different primers have different ideal conditions, and without control over individual reactions, bias might be introduced. In Ferreira et al., 2023, qPCR was used. Was it used here as well? Even so, the previous study was only concerned with *Eimeria*, so how did the authors in the current study account for multiple kingdoms of life? For example, in Fig 2A,B the data from eukaryotes is lumped together, but given it spans 18S, 28S and ITS, I'm not sure that this is appropriate (see comments on Figure 2).

Data subsetting: From the code it seems that parasites and fungi were subset down to select taxa, while bacteria were not. What was the rationale? Isn't the goal to get a wholistic view of the microbial community? If the data was not normalized and then certain clades were subset even further, I'm not sure it is appropriate to compare across groups. For example, one main claim is that bacteria had a stronger association on immune measures than parasites and fungi, but perhaps that is because the entire bacteriome was considered versus just a subset. Line 301 says data was subset to "known parasites"- what about other microeukaryotes that might correlated with ill health? Also, how were there pathogens selected? *Toxoplasma*, a known hyena

parasite, was left off this list but was detected in the study. What about microeukaryotes that may have beneficial roles (see: <https://www.frontiersin.org/journals/microbiology/articles/10.3389/fmicb.2023.1123513/full>)?

Sample metadata: It's unclear how many runs were performed or whether data was analyzed for batch effect. The metadata should reflect which run each sample came from as well as the negative controls. DNA concentrations used in Decontam (are these from Qubit or qPCR?) should be included as well. Were technical replicates included (i.e., same sample, different run/reaction)? If so, these should be specified in the metadata (even if they weren't all used in the main analysis)

Confounding age & sex covariates: Why were only adult females sampled? The rationale here should be described. Was it not possible to sample adult males? From Ferreira et al., 2021: "*Fecal concentrations of IgA, IgG, and mucin increased with Ancylostoma egg load and were higher in juveniles than in adults. Females had higher mucin concentrations than males*". The fact that all adults in this study are female makes it more difficult to parse out the influence of sex, age, and immune measures on the microbiome.

FIGURES: In general, I felt that there was redundancy in the figures, while some important concepts were not visualized. Specifically, Table 1 conveys the same information as Figs 2E,F and Fig 3A-C. The goal of the paper is to show an association between immune measures and microbial diversity, yet we don't see plots of alpha and beta diversity measures over IgA/mucin levels aside from the pairwise comparisons (which can be hard to digest). In addition, although temporal distance between samples is significant for all microbial groups in Table 1, only age is further analyzed.

▪ **FIGURE 1:**

- I don't find the visualizations of the pairwise distances in B,D and F that interesting, they could go in the supplemental. Instead, I'd rather see the histograms for the clans, social ranks and year of sampling from Fig S1 since these are the covariates used in the model in Table 1.
- I liked the hyena icons at first, but now feel like they are distracting. This is especially true for 1B since according to the dyad.data.rds there aren't any adult pairs at Age distance=0 (only juvenile-juvenile pairs). I think it's best to just note "more similar in age" or "similar f-IgA levels" with an arrow pointing towards "dissimilar ages" or something along those lines...
 - **Legend:** The abbreviation RU is defined, yet I do not see it used in the figure. I believe "immune measures levels" should be "immune measure levels".
 - **1A:** This should be stretched out or divided into hyenas sampled once and those with repeat sampling because right now it is hard to see what is happening up top.
 - **1B:** I'd label the x axis "Pairwise age distances"

- **1D**: Incorrectly labeled 1E in legend. I'd label the x axis "Pairwise f-IgA distances" or "dissimilarities"
 - **1E**: Incorrectly labeled 1D in legend
- **TABLE 1**: First off, I'd start with the relative abundance plots (Fig 2A-D) and then show how these communities are associated with immune measures. If using this table (instead of Figs 2E,F & Fig 3A-C), I'd clean it up a bit by making it more obvious which are significant (like the supplemental table) and adding labels on the side to delineate the immune measures, social/environmental covariates and interactions. Shouldn't a sex:mucin interaction be added given previous findings (see above)? Perhaps you could repeat the analysis on just juveniles to investigate sex.
 - **TABLE 1 and Figs 2E,F & Fig 3A-C**: How does one interpret these results in plain terms? Most tables like this list the covariates as predictors, not the pairwise distance between covariates. And is there directionality associated with +/- values? That should be clearly labeled. The figures are also confusing because the x-axis is labeled with the covariate distance, but I believe it should be "BRM Estimate size". For example: Fig 2E has the x-axis title "f-IgA distance" yet the values are all negative and the f-IgA distances shown in 1D are all positive.
 - **FIGURE 2**: Fig 2A-D feel like one figure about relative abundance, while E,F pertain more to Table 1.
 - **2A,B**: Again, for the reasons discussed above I don't think it is appropriate to plot these microbial groups together. This is especially true in B where the relative abundance adds up to 100% (which is arbitrary). At least split these up into parasites and fungi.
 - **2E**: These values (Overall, Parasite) don't match the table. Your code does provide the correct version, however.
 - **2E,F**: If you keep these I'd make the x-axes consistent so the differences are more clear.
 - **FIGURE 3**:
 - **3A-C**: I prefer table 1 since it is more obvious which are significant. If keeping these plots, I'd lighten the bars that cross 0 to highlight the significant groups
 - **3D,E**: Can you clarify what was done here? Did you simply take the model from Table 1 and enter either AgeDistance=0 or 1 and then generate predictions along set intervals from f-IgA 0 to 1 (similar to dissimilar pairwise f-IgA comparisons?). This is hard to interpret without the number of samples and individuals used. Again, there are no adult-adult with AgeDistance=0 in the dyad.data, so 0 is only juveniles (both are shown in the legend). Also, there is only one pairwise comparison where AgeDistance=0 (sample P816_B8320 and M148_X5461). I'm not sure this is the best way to present the data. We already know from previous research that the microbiome of adults and young juveniles should look

different and that IgA/mucin is associated with age in hyenas. I'd prefer a visualization of your data than model predictions. I think it would be more effective to stratify by age groups and look for associations within each. Perhaps just ordinate the microbial diversity for juveniles, see which axis separates the data by f-IgA and plot the axis loadings over f-IgA (then repeat for adults).

- **Longitudinal analyses:** I found it odd that even though temporal distance (time) was found to be significantly influential for all microbial groups tested, this data wasn't visualized. Was seasonality explored at all? I'd expect it might affect parasite load. I'd love to see visualizations showing microbial diversity metrics (α diversity and beta diversity ordination loadings), the abundance of microbial groups, and immune measure levels over age AND over time.
 - How were longitudinal samples handled? Including multiple samples from the same animals could inflate the association between the microbiome and immune measures. These longitudinal samples could also be used to analyze within individual changes over time (especially the individuals with 3 samples) although this may be outside the scope of this analysis.
- **FIGURE 4:** See above for comments related to the host & unclassified DNA...
 - **4A,C:** I don't find these graphs very interesting aside from the R² value, they could be supplemental. What is the shading? Are these credible or confidence intervals?
 - **4B,D:** These plots are more interesting but don't give information on directionality. I'd suggest that rather than sorting by importance, you provide a heatmap of samples sorted by f-IgA or mucin measurement levels and the relative abundance of these taxa
 - **4E:** I like the use of a network but would like to see it without host DNA. Also, please keep the colors chosen for your microbe groups consistent throughout the figures.

Minor comments:

- **Title:** "Symbiont" doesn't seem appropriate here, especially since the data selection for the eukaryotes was geared towards parasites. Also be careful of implying causation.
- **Line 57:** "Intestinal mucosa" often refers to the GI layer (i.e., the epithelium, lamina propria & muscularis). Did you mean the "intestinal mucosal epithelium" of the small intestines/colon (remember that distal GI tract regions have stratified epithelium). Keep in mind which regions fecal samples likely represent.
- **Line 61:** Recommend "can be associated" since perturbations can also lead to a new baseline community that is not necessarily unhealthy (e.g., a diet shift)
- **Line 78:** "inter-kingdom"?

- **Line 83:** Over-generalization. This sentence refers to “wild animals” in captivity- in the context of this study, readers may assume “captivity” means zoos. However, two referenced studies (27, 29) are on mice (I’d argue lab mice have unique husbandry) and the other sampled humans (28). The latter included published data on wild/captive mammals, yet wasn’t designed to explore captivity. Ley found that host phylogeny/diet overshadowed other covariates. More recent studies of wild mammal bacteriomes suggest that certain host clades (e.g., Perissodactyls) are resistant to the captivity-induced changes (see doi: 10.3389/fevo.2021.785089 and DOI: 10.1093/icb/icx090). Suggest rewording and/or adjusting references.
- **Line 87:** Flow is a bit confusing. Line 80 gives a nice transition to wild mammals, but then 83 refers to mice/humans and then 87 is vague (yet the refs have transitioned to studies of wild animals). IMO, this paragraph should focus on the growing knowledge surrounding wild mammalian population microbiomes, the feasibility of non-invasive sampling and the implications for conservation or understanding microbiome-environmental interactions.
- **Lines 106-111:** Really nice job stating the hypothesis and tying in the need for mixed community microbiome analysis using refs to previous microbiome work and the *Anycyclostoma*/mortality example! I would try to flesh out the sentence about the previous microbiome, parasite and immune measure studies that were recently done (I believe by your lab) to make it clear what findings informed your current work and how this study helps complete the story.
- **Lines 115-125:** Impressive dataset and I appreciate the use of host genotyping. Is this the only sentence referring to the host genotyping methods? The reference is good, but you might want to say whether this leveraged an existing hyena reference database (and if so, mention the database coverage in terms of the current population). Perhaps mention in the supplemental methods.
- **Lines 126 (and 241 etc...):** In general, figures presenting actual data should be called out in the results.
- **Line 138:** I’d expand on the sample collection/field conditions. How were the samples collected immediately (without risk to researchers)? How was anthropogenic manipulation (Line 95) minimized in this study? Were cool packs in a cooler? Approximately how long did samples sit before reaching the field station?
- **Line 148:** Suggest “Microbial DNA extraction” to differentiate from the host genotyping?

- **Line 149:** Perhaps specify what the NucleoSpin Soil kit targets. I was unfamiliar and the Takara blurb only mentions bacteria, yeast, and fungi. Maybe include a reference for its use in helminths (e.g., doi: [10.1186/s13028-022-00624-3](https://doi.org/10.1186/s13028-022-00624-3))
- **Line 215:** I interpret “repeatability” to mean the comparison of one sample repeatedly sequenced under equivalent conditions (technical replicates, e.g., PCR duplicates). Are you referring to technical replicates? It seems more likely that you are comparing longitudinal samples from hyenas that were sampled over time. This should be clarified given that these are very different research questions.
- **Lines 221-222:** I interpret “dyadic” to mean two linked individuals, and in this case, one could take it to mean mother-pup dyads. Do you mean “pairwise” here? State the specific beta diversity measure used (instead of “(distances)”). I’d reword or remove “(excluding within-sample comparisons)” since for a moment I thought you were referring to alpha diversity (but I think you mean comparing a sample to itself which would have BC=0)?
- **Line 228:** Will all readers be familiar with Stan? Perhaps say “in the Stan probabilistic programming language” and list the R interface used (RStan, cmdstanr?) + version
- **Line 232:** Looks like “the Sampler” should be lower-case when used as such https://cran.r-project.org/web/packages/brms/vignettes/brms_overview.pdf
- **Line 244:** Did you test for all interactions in the model before arriving at the final version? The reference you provided did find interactions between age and immune measures, but since here you are testing for influence of immune measures/environmental/social covariates on microbial diversity, you may reveal different interactions. Glancing at the code it appears that the immune/age interactions were manually entered but I don’t see that the other variables were tested for interactions (i.e., separated via “+” rather than “*”)
- **Lines 248-250:** Recommend shortening this section by adding `splitrule=variance` and `min.node.size=10` to the previous sentence (since they were the same for both immune measures) and then just specify the `mtry` for each
- **Line 253:** Describing taxa importance as permutation importance is a bit redundant and doesn’t tell me much. Perhaps specify that this is a command argument (`importance="permutation"`) and explain what permutation importance means (something like “a variable’s influence on the model’s predictive performance”)
- **Line 253:** Give the mean and range of samples collected for those 37 hyenas

- **Lines 298-300:** Contamination/final dataset. I realized contamination should be minimized due to the microfluidics approach but what was the percent of reads/ASVs removed by Decontam? Comment briefly on the ASV/sample drop out post-QC. On that note, 301 bacteria + 691 eukaryotes= 992 cASVs (not 999). Among eukaryotes 420 were fungi and 25 were known pathogens, what about the other 246 cASVs?